# Latent-Hysteresis Graph ODEs:
## Modeling Coupled Topology-Feature Evolution via Continuous Phase Transitions

## Abstract

Graph neural ordinary differential equations (Graph ODEs) make a shift in learning on graphs from discrete layers to continuous time steps, transforming information flow on graphs into a smooth and adaptive process. However, we show that Graph ODEs with strictly positive and irreducible mixing operators will lead to an inherent *monostability trap*: in the infinite-depth evolution, there will be unavoidable information leakage and the dynamics will converge to a single global attractor (*i.e.*, consensus).

To overcome this limitation, we propose the **Hysteresis Graph ODE (HGODE)**, where feature and topology achieve coupled evolution and are connected by a learned force between nodes. By maintaining a latent continuous potential field and bipolarized gating, the edge state alternates between open and insulated states. By this approach, the model achieves a phase transition without hurting the differentiability. We provide theoretical guarantees for the asymptotic behavior of our model, as well as the empirical validation for its performance on synthetic and real-world datasets.

## 1. Introduction

Graph neural network (GNN) is a dominant paradigm for relational learning (Zhou et al., 2018; Wu et al., 2020), but its inherent discrete update across layers limits the ability on continuous modelling. To mitigate the gap between discrete and continuous dynamics, Neural ordinary differential equations (NODEs) (Chen et al., 2018) extend parameter update from discrete layer to (almost) continuous time steps. Building on this idea, Graph ODEs (GODEs) (Poli et al., 2019; Liu et al., 2025) model node representations as a continuous flow over a graph, which in principle supports long-range dependency modeling beyond finite-depth message passing.

Despite this promise, we show that a broad class of Graph ODEs admits an intrinsic feature collapse risk in the long-time regime. Specifically, when the propagation operator is row-stochastic and remains strictly positive on a strongly connected support, the induced mixing is irreducible, and the dynamics converge to a unique global attractor. This condition is satisfied by many dense soft-weight constructions (*e.g.*, temperature-scaled global attention or dense similarity kernels), and it implies unavoidable information leakage and over-smoothing as time $t \to \infty$.

We address this limitation by explicitly modeling topology as a dynamical state. We propose the **Hysteresis Graph ODE (HGODE)**, which augments feature evolution with a co-evolving latent topological potential governed by bistable hysteresis dynamics. A learned pairwise force drives each edge potential within a non-convex double-well landscape; the resulting potential barrier induces hysteresis and endows the topology with structural memory. This coupled system enables differentiable phase transitions that polarize edges into connected versus insulated states, thereby breaking irreducible mixing. Consequently, HGODE fundamentally changes the asymptotic behavior of continuous-depth propagation, allowing convergence to cluster-wise invariant manifolds rather than a single global consensus.

Our main contributions are:

- We characterize the long-time behavior of diffusion-style Graph ODEs under strictly positive row-stochastic mixing, and extend the consensus-collapse result to a class of time-varying operators satisfying uniform positivity.

- We introduce a coupled feature–topology ODE where a double-well edge potential and learned force induce topological phase transitions, explicitly breaking irreducibility and preventing global consensus collapse.

- We connect the theory to training via a force-margin objective aligned with the hysteresis threshold, and validate the performance on theory-driven diagnostics and real-world benchmarks.

[1]Anonymous Institution, Anonymous City, Anonymous Region, Anonymous Country. Correspondence to: Anonymous Author <anon.email@domain.com>.

Preliminary work. Under review by the International Conference on Machine Learning (ICML). Do not distribute.

## 2. Related Work

**Graph neural ODEs and continuous-depth GNNs.** Graph neural ODEs cast message passing as continuous-time dynamics by defining node representations as solutions of differential equations on graphs. Representative formulations such as GDE (Poli et al., 2019) and GRAND (Chamberlain et al., 2021) model propagation through diffusion-like dynamics (possibly with stochasticity), offering a principled view of infinite-depth architectures and adaptive computation. Earlier existing designs evolve features on a fixed topology or on dense soft-weight operators, which remain strictly positive on a strongly connected support in common instantiations (Bodnar et al., 2022; Rusch et al., 2022; Maskey et al., 2024). Recent advancements have expanded Graph ODEs beyond simple diffusion to address complex evolution dynamics and over-smoothing. CSG-ODE (Wang et al., 2025b) introduces a "ControlSynth" mechanism to model dynamic graphs, focusing on temporal evolution and trajectory prediction. SEGNO (Liu et al., 2024) and DuSEGO (Wang et al., 2025a) employ second-order ODEs to incorporate inertial inductive biases and equivariance, which naturally mitigate over-smoothing by maintaining particle momentum. In contrast, our work treats topology as a dynamical state and focuses explicitly on the long-time regime where the induced mixing structure determines whether collapse is inevitable.

**Over-smoothing and long-time feature collapse.** Over-smoothing has been extensively studied in discrete GNNs, where repeated aggregation drives node representations toward low-frequency subspaces or consensus, and consequently impair discrimination (Rusch et al., 2023; Wu et al., 2023b;c; Hou et al., 2025). Theoretical analyses have rigorously linked this phenomenon to the spectral contraction of the graph Laplacian and the convergence of random-walks to a stationary distribution (Oono & Suzuki, 2020). Standard mitigation strategies such as residual connections, normalization layers (Zhao & Akoglu, 2020), and personalized diffusion (Gasteiger et al., 2018; Min et al., 2020; Koishekenov, 2023), have proven effective in finite-depth regimes. However, these approaches generally modify feature evolution over a fixed topology. Consequently, they do not explicitly control the fundamental asymptotic mixing properties of the operator itself (Chamberlain et al., 2021), leaving continuous-depth models susceptible to inevitable collapse in the infinite-time limit.

**Graph structure learning and adaptive adjacency.** A large body of work learns or refines graph structure via adaptive adjacency matrices, metric-based graphs, and attention mechanisms that assign edge weights from features (Franceschi et al., 2019; Chen et al., 2020b; Wang et al., 2019; Zheng et al., 2024). Soft attention can be viewed as a continuous form of structure learning, yet it is typically dense and strictly positive, and thus often preserves global mixing in the long-time regime (Lee et al., 2019; Velickovic et al., 2018; Ye & Ji, 2021). Other approaches learn static latent graphs (Gasteiger et al., 2019; 2018) or perform graph denoising prior to propagation (Jin et al., 2020), but do not treat topology as a coupled dynamical variable. In contrast, HGODE introduces a latent topological potential that co-evolves with features and admits bistable states, enabling persistent edge polarization and structural memory.

## 3. Monostability Trap and Hysteretic Topology Dynamics

### 3.1. Notations

For a given graph $\mathcal{G} = \{\mathcal{V}, \mathcal{E}\}$, we represent $\mathcal{V} = \{v_1, v_2, \ldots, v_N\}$ and $\mathcal{E} = \{e_{ij} \mid v_i, v_j \in \mathcal{V}\}$ as the node set and edge set, respectively. For each node $v_i$, we define a $m$-dimensional feature vector $\mathbf{h}_i \in \mathbb{R}^m$, forming the global feature matrix $\mathbf{H} \in \mathbb{R}^{N \times m}$.

Since our framework allows for asymmetric interactions, we define the degree matrix $\mathbf{D}(t) = \text{diag}(d_1(t), \ldots, d_N(t))$ with $d_i(t) = \sum_j \widetilde{A}_{ij}(t)$, where $\widetilde{\mathbf{A}}(t) = \mathbf{A}(t) + \epsilon \mathbf{I}$ adds a small self-loop for numerical stability. The diffusion is governed by the row-stochastic matrix $\mathbf{P}(t) = \mathbf{D}(t)^{-1}\widetilde{\mathbf{A}}(t)$. The diffusion dynamics are governed by the row-stochastic transition matrix $\mathbf{P}(t) = \mathbf{D}(t)^{-1}\mathbf{A}(t)$. For any weighted adjacency matrix $\mathbf{A}$, we denote its topological support as $\mathcal{E}_{supp} := \{(i,j) : \mathbf{A}_{ij} > 0\}$.

For the proposed framework, we introduce a continuous latent potential matrix $\mathbf{U} \in \mathbb{R}^{N \times N}$. To ensure scalability, we restrict computations to a sparse candidate set $\mathcal{E}_{cand} \subset \mathcal{V} \times \mathcal{V}$ constructed once at initialization (*e.g.*, 2-hop, Laplacian random-walk, and random neighbors), and define the effective active set $\mathcal{E}_{active}(t) \subseteq \mathcal{E}_{cand}$ by filtering edges whose effective weights become negligible during annealing. We denote the topological force function as $\mathcal{F} : \mathbb{R}^m \times \mathbb{R}^m \to \mathbb{R}$, and the structural annealing parameter as $\mu(t)$. Finally, $\tau_{feat}$ and $\tau_{topo}$ denote the timescale constants for feature and topological evolution, respectively.

All of the proofs can be found in the Appendix A.

### 3.2. The Monostability of Directed Consensus

We analyze the asymptotic behavior of continuous-time GNNs. Unlike previous works that rely on undirected Dirichlet energy, we consider the general case of directed, non-symmetric interactions characteristic of global soft-attention mechanisms. Our analysis is grounded in *consensus dynamics* on graphs with a strictly positive weighted topology, as induced by the propagation operator.

**Definition 3.1** (Directed Diffusion Dynamics). Let $\mathbf{P} = \mathbf{D}^{-1}\mathbf{A}$ be a (time-invariant) row-stochastic transition ma-

trix of the graph. The evolution of node features $\mathbf{H}(t)$ is governed by the consensus flow:

$$\frac{d\mathbf{H}}{dt} = -(\mathbf{I} - \mathbf{P})\mathbf{H} \qquad (1)$$

which describes a relaxation process where each node continuously migrates towards the convex hull of its neighbors.

Note that while we begin with the time-invariant case for clarity, we later extend the analysis to time-varying mixing operators, which are more representative of attention-based Graph ODEs used in practice. We then give the assumption on the topology matrix:

**Assumption 3.2** (Strictly Positive Mixing on a Strongly Connected Support). Recall that $\mathcal{E}_{supp} := \{(i,j) : A_{ij} > 0\}$ is the support of the propagation operator. We assume that (i) $A_{ij} > 0$ on $\mathcal{E}_{supp}$ and (ii) the directed graph $(\mathcal{V}, \mathcal{E}_{supp})$ is strongly connected. Equivalently, the row-stochastic matrix $\mathbf{P} = \mathbf{D}^{-1}\mathbf{A}$ is irreducible.

This assumption is satisfied by many dense soft-weight constructions (*e.g.*, global attention or dense similarity kernels). Explicit hard masking/sparsification can break irreducibility, but such mechanisms are external to the continuous dynamics and do not address the long-time behavior in a principled manner. Under Assumption 3.2, we invoke the Perron-Frobenius theorem to derive the inevitability of over-smoothing.

**Theorem 3.3** (Consensus Trap under Irreducible Positive Mixing). *Under Assumption 3.2, the Markov matrix* $\mathbf{P}$ *admits a unique stationary distribution* $\boldsymbol{\pi} \succ 0$ *such that* $\boldsymbol{\pi}^\top \mathbf{P} = \boldsymbol{\pi}^\top$ *and* $\boldsymbol{\pi}^\top \mathbf{1} = 1$. *Moreover, the directed consensus flow* $\frac{d\mathbf{H}}{dt} = -(\mathbf{I} - \mathbf{P})\mathbf{H}$ *converges to the rank-one consensus subspace:*

$$\lim_{t\to\infty} \mathbf{H}(t) = \mathbf{1}\big(\boldsymbol{\pi}^\top \mathbf{H}(0)\big). \qquad (2)$$

*The convergence rate is exponential and is governed by the spectral gap of* $(\mathbf{I} - \mathbf{P})$, *i.e., by* $-\max_{k\geq 2} \operatorname{Re}(\lambda_k(\mathbf{P}) - 1)$.

Theorem 3.3 shows that irreducible positive mixing forces the dynamics into a single global attractor, hence long-time information leakage is unavoidable. Escaping this trap requires altering the asymptotic mixing structure—most notably, breaking irreducibility so that the effective propagation becomes reducible (*e.g.*, block-diagonal up to permutation), which yields multiple invariant subspaces corresponding to different clusters.

### 3.3. Beyond time-invariant mixing

Theorem 3.3 establishes the monostability trap for time-invariant propagation operators. However, many diffusion-style Graph ODEs induce mixing operators that vary with time through their dependence on node features. We next show that the consensus trap persists under a broad class of time-varying row-stochastic operators. We start from an assumption for time-varying topology $\mathbf{P}(t)$:

**Assumption 3.4** (Uniform Positivity). Let $\mathbf{P}(t)$ be a time-varying row-stochastic matrix. We assume that there exists a constant $\alpha > 0$ (necessarily $\alpha \leq 1/N$) such that: (i) $\mathbf{P}(t)$ is piecewise continuous in $t$; (ii) (Uniform Positivity) $\mathbf{P}_{ij}(t) \geq \alpha$ for all $i,j \in \{1, \ldots, N\}$ and all $t \geq 0$.

**Lemma 3.5** (Window Contraction). *Under Assumption 3.4, there exists a contraction rate* $\rho \in (0,1)$ *such that for any feature signal* $\mathbf{x}(t) \in \mathbb{R}^N$ *governed by* $\dot{\mathbf{x}}(t) = -(\mathbf{I} - \mathbf{P}(t))\mathbf{x}(t)$, *the following holds:*

$$\operatorname{diam}(\mathbf{x}(t + T_w)) \leq (1 - \rho)\operatorname{diam}(\mathbf{x}(t)), \quad \forall t \geq 0,$$

*where* $\operatorname{diam}(\mathbf{x}) := \max_i x_i - \min_i x_i$.

Then, we give the theorem for time-varying mixing:

**Theorem 3.6** (Consensus trap under time-varying positive mixing). *Consider the time-varying consensus flow*

$$\frac{d\boldsymbol{H}(t)}{dt} = -(\boldsymbol{I} - \boldsymbol{P}(t))\boldsymbol{H}(t),$$

*where* $P(t)$ *satisfies Assumption 3.4. Then the dynamics converge exponentially to the rank-one consensus subspace. Specifically, for each feature dimension* $k$, *there exist constants* $C, \kappa > 0$ *such that*

$$\operatorname{diam}(\boldsymbol{H}_{.k}(t)) \leq Ce^{-\kappa t} \operatorname{diam}(\boldsymbol{H}_{.k}(0)).$$

*Equivalently, there exists* $y(t) \in \mathbb{R}^m$ *such that*

$$\lim_{t\to\infty} \|\boldsymbol{H}(t) - \mathbf{1}y(t)^\top\|_F = 0.$$

Naturally, while defining $\mathbf{P}$ as the soft-attention, we have:

**Corollary 3.7** (Dense soft-attention induces the consensus trap). *Let* $P(t)$ *be defined by fully-connected softmax attention,*

$$\boldsymbol{P}_{ij}(t) := \frac{\exp(\langle \boldsymbol{h}_i(t), \boldsymbol{h}_j(t)\rangle/\tau)}{\sum_k \exp(\langle \boldsymbol{h}_i(t), \boldsymbol{h}_k(t)\rangle/\tau)}.$$

*Assume that node features remain bounded, i.e.,* $\|h_i(t)\| \leq B$ *for all* $i, t$. *Then* $\boldsymbol{P}(t)$ *satisfies Assumption 3.4 with*

$$\boldsymbol{P}_{ij}(t) \geq \frac{1}{N}\exp\Big(-\frac{2B^2}{\tau}\Big).$$

*Consequently, attention-based Graph ODEs with dense soft mixing inevitably converge to the consensus subspace in the long-time regime.*

The boundedness assumption is mild in practice and is typically supported by architectural normalization, regularization, and the stability of numerical ODE solvers. Increasing the attention temperature $\tau$ enlarges the uniform positivity

*Figure 1.* **Hysteresis Graph ODE (HGODE) controls asymptotic mixing.** (A) **Monostability trap:** diffusion-style Graph ODEs with uniformly positive (including time-varying) mixing collapse to a single consensus attractor in the long-time regime. (B) **Coupled dynamics:** HGODE jointly evolves node features and latent edge potentials, treating topology as a dynamical state. (C) **Hysteresis:** a double-well landscape with a critical threshold induces bistability and structural memory, polarizing edges into connected versus insulated phases. (D) **Force–potential polarization:** the learned force $\mathcal{F}$ drives edge potentials $\mathbf{U}$ into connected or insulated phases. (E) **Asymptotic outcome:** edge polarization breaks irreducible mixing, yielding cluster-wise invariant subspaces (*e.g.*, multiple near-zero Laplacian eigenvalues) instead of global consensus.

constant, leading to faster convergence to consensus. Nevertheless, this result targets dense soft mixing (full support). If explicit sparsification/masking is applied, uniform positivity may fail and the long-time behavior depends on the induced reducible support.

### 3.4. Hysteresis via Double-Well Potentials

To prevent collapse to a single attracting basin, the topology dynamics must be *bistable*. We model each directed edge $e_{ij}$ via a latent order parameter $u_{ij}(t)$ evolving on a non-convex Landau energy landscape with two stable structural phases: *Connected* and *Insulated*, rather than as a directly optimized propagation weight.

**Definition 3.8** (Landau Double-Well Potential for Connectivity). Let $u_{ij} \in \mathbb{R}$ be the latent order parameter associated with the directed edge $e_{ij}$. Given the node features $\mathbf{h}_i, \mathbf{h}_j$, we define the driving force $\mathcal{F}_{ij}(t) := \mathcal{F}(\mathbf{h}_i(t), \mathbf{h}_j(t))$. We define the local Landau potential (energy) for each edge as

$$V(u_{ij}; \mathcal{F}_{ij}) := \underbrace{\frac{1}{4}u_{ij}^4 - \frac{1}{2}u_{ij}^2}_{\text{intrinsic bistability}} - \underbrace{\mathcal{F}_{ij}\, u_{ij}}_{\text{coupling field}}, \quad (3)$$

where the intrinsic term yields two degenerate minima at

$u_{ij} = \pm 1$ in the absence of external drive ($\mathcal{F}_{ij} = 0$), and the topological force $\mathcal{F}_{ij}$ acts as an external bias that breaks the symmetry.

**Gradient-flow dynamics.** We evolve the edge state by gradient descent on the potential:

$$\frac{du_{ij}}{dt} = -\frac{\partial V(u_{ij}; \mathcal{F}_{ij})}{\partial u_{ij}} = u_{ij} - u_{ij}^3 + \mathcal{F}_{ij}. \quad (4)$$

The cubic nonlinearity in (4) induces **hysteresis**: the state of an edge depends not only on the instantaneous force $\mathcal{F}_{ij}$ but also on its history (*i.e.*, which well it currently occupies), thereby providing a form of structural memory.

**Proposition 3.9** (Bistability, saddle-node transitions, and hysteresis). *For each edge $(i, j)$, the equilibria of (4) are given by the roots of:*

$$u_{ij}^3 - u_{ij} = \mathcal{F}_{ij} \quad (5)$$

*Define the critical force magnitude:*

$$\mathcal{F}_{\text{crit}} := \frac{2}{3\sqrt{3}}. \quad (6)$$

*Then the system undergoes* saddle-node (fold) bifurcations *at $\mathcal{F}_{ij} = \pm\mathcal{F}_{\text{crit}}$, with the following regimes:*

1. **Bistable regime** ($|\mathcal{F}_{ij}| < \mathcal{F}_{\text{crit}}$). *Eq. 5 admits two stable equilibria (attractors) and one unstable equilibrium (repeller). Consequently, an edge initialized in the* Connected *well ($u_{ij} \approx +1$) resists switching to* Insulated *even if $\mathcal{F}_{ij}$ decreases moderately, and vice versa, yielding robustness to noise and transient perturbations.*

2. **Monostable regime** ($|\mathcal{F}_{ij}| > \mathcal{F}_{\text{crit}}$). *Only one real fixed point remains. The barrier disappears via a saddle-node transition, forcing a rapid, deterministic phase change to the single remaining attractor.*

**From latent states to effective propagation.** In our model, the effective propagation weight $\mathbf{A}_{ij}(t)$ is obtained via a projection $\phi : \mathbb{R} \to [0, 1]$ of the latent state $u_{ij}(t)$. Specifically, we employ a shifted sigmoid or projection function (*e.g.*, $\phi(u) = \sigma(u/\tau_{gate})$) such that the *Connected* well ($u \approx +1$) maps to high attention weights, while the *Insulated* well ($u \approx -1$) is effectively suppressed to zero ($\mathbf{A}_{ij} \approx 0$). This weight is further pruned by structural annealing on the static candidate set $\mathcal{E}_{cand}$. Since near the transition region $|\mathcal{F}_{ij}| \approx \mathcal{F}_{\text{crit}}$, the vector field can change rapidly due to saddle-node transitions, which makes the coupled dynamics numerically sensitive. We therefore use adaptive ODE solvers (`dopri5`) for stable integration; fixed-step solvers (Euler/RK4) may require prohibitively small step sizes to accurately resolve switching events.

**From latent state to propagation weight.** We convert each latent potential $u_{ij}(t)$ into an effective propagation weight $A_{ij}(t) \in [0, 1]$ using a monotone gate (*e.g.*, a temperature-controlled sigmoid), so that the Connected well maps to high weight while the Insulated well maps to near-zero weight (see Sec. 4.1 for the formal definition).

# 4. The Hysteresis Graph ODE Framework

We now instantiate the coupled feature–topology ODE that enforces the reducible/clustered mixing structure implied by Sec. 3. The overview of the proposed HGODE is showed in Figure 1. We illustrate the details in the following sections.

## 4.1. Coupled Evolutionary Dynamics

HGODE governs the co-evolution of node features $\mathbf{H}(t)$ and latent edge potentials $\mathbf{U}(t)$ on a *static* sparse candidate set $\mathcal{E}_{cand}$. The coupled dynamics are defined as:

$$
\begin{cases}
\tau_{feat} \dfrac{d\mathbf{H}(t)}{dt} = \mathcal{G}_\phi(\mathbf{H}(t), \mathbf{A}(t)) - \gamma \, \mathbf{H}(t), \\[2mm]
\tau_{topo} \dfrac{d\mathbf{U}(t)}{dt} = (1 - \lambda) \, \mathbf{U}(t) - \mathbf{U}(t)^3 + \mathcal{F}_\theta(\mathbf{H}(t)),
\end{cases}
\tag{7}
$$

where $\mathcal{G}_\phi$ is a graph neural operator, $\gamma \geq 0$ is a feature decay coefficient, and $\lambda \in [0, 1)$ controls the depth of the double-well potential.

**Force field.** $\mathcal{F}_\theta(\mathbf{H}(t)) \in \mathbb{R}^{N \times N}$ is a matrix-valued force field, whose entries depend on feature compatibility:

$$
\big(\mathcal{F}_\theta(\mathbf{H})\big)_{ij} = \mathcal{F}_\theta(\mathbf{h}_i, \mathbf{h}_j).
$$

Its parameterization and the margin-inducing training objective are specified in Sec. 4.3.

**From latent potentials to effective propagation.** The effective propagation weights are obtained by a gated mapping of $\mathbf{U}(t)$ and restricted to the static candidate set:

$$
\mathbf{A}_{ij}(t) = \sigma\left(\frac{\mathbf{U}_{ij}(t)}{\tau}\right) \cdot \mu(t) \cdot \mathbf{1}\big[(i, j) \in \mathcal{E}_{cand}\big], \tag{8}
$$

where $\sigma(\cdot)$ is the sigmoid gate with temperature $\tau > 0$, and $\mu(t)$ is the structural annealing schedule used for filtering negligible connections). Importantly, the state dimension is fixed during integration since $\mathbf{U}(t)$ evolves only on $\mathcal{E}_{cand}$.

## 4.2. Coupled Conditional Gradient Flows

Eq. (7) admits an interpretation as *coupled conditional gradient flows*:

$$
\begin{cases}
\tau_{feat} \dfrac{\partial \mathbf{H}}{\partial t} = -\nabla_{\mathbf{H}} \mathcal{E}_{feat}(\mathbf{H}; \mathbf{U}) \\[2mm]
\tau_{topo} \dfrac{\partial \mathbf{U}}{\partial t} = -\nabla_{\mathbf{U}} \mathcal{E}_{topo}(\mathbf{U}; \mathbf{H})
\end{cases}
\tag{9}
$$

where $E_{\text{feat}}$ and $E_{\text{topo}}$ are two interacting energy landscapes. This viewpoint mainly serves as a compact way to expose the implicit objectives optimized by the coupled feature-topology dynamics.

**Topology energy.** Conditioned on $H$, the topology dynamics in (7) correspond to gradient descent on a separable Landau-type energy over $(i, j) \in \mathcal{E}_{\text{cand}}$:

$$
\sum_{(i,j) \in \mathcal{E}_{cand}} \left[ \frac{1}{4} \mathbf{U}_{ij}^4 - \frac{1 - \lambda}{2} \mathbf{U}_{ij}^2 - \big(\mathcal{F}_\theta(\mathbf{h}_i, \mathbf{h}_j)\big) \mathbf{U}_{ij} \right]
\tag{10}
$$

This matches the double-well mechanism analyzed in Sec. 3.4, with $\lambda$ controlling the barrier height and $F_\theta$ acting as the feature-conditioned bias.

**Feature energy (diffusion-style instantiation).** When $\mathcal{G}_\phi$ is instantiated as a diffusion operator, *e.g.*, $\mathcal{G}_\phi(\mathbf{H}, \mathbf{A}) = \mathbf{P}\mathbf{H} - \mathbf{H}$ with $\mathbf{P} = \mathbf{D}^{-1}\mathbf{A}$, the feature dynamics correspond to descent on a regularized Dirichlet-type energy:

$$
\mathcal{E}_{feat}(\mathbf{H}; \mathbf{U}) = \frac{1}{2} \text{Tr}\big(\mathbf{H}^\top (\mathbf{I} - \mathbf{P})\mathbf{H}\big) + \frac{\gamma}{2} \|\mathbf{H}\|_F^2. \tag{11}
$$

This drives local consensus *within* connected components induced by the effective topology. More general message-passing operators can be viewed as parameterized generalizations of this diffusion archetype.

### 4.3. Topological Force Field and Scaling

The topological force $\mathcal{F}_\theta$ is the key interface that couples features to topology. It determines whether an edge remains in its current well or crosses the hysteresis threshold $\mathcal{F}_{\text{crit}}$ (Sec. 3.4) to switch phase.

We parameterize $\mathcal{F}_\theta$ as a bounded edge-wise score computed from concatenated node features:

$$\mathcal{F}_{ij} := \big(\mathcal{F}_\theta(\mathbf{H})\big)_{ij} = s \cdot \tanh\Big(\text{MLP}_\theta\big([\mathbf{h}_i\|\mathbf{h}_j]\big)\Big) \quad (12)$$

where $(i,j) \in \mathcal{E}_{cand}$. $\tanh(\cdot)$ ensures $\mathcal{F}_{ij} \in [-s, s]$ and $s$ is a scale factor chosen such that $s \gtrsim \mathcal{F}_{\text{crit}}$.

The sign of $\mathcal{F}_{ij}$ biases the edge state toward the *connected* ($u_{ij} > 0$) or *insulated* ($u_{ij} < 0$) well, while the magnitude controls whether switching becomes deterministic ($|\mathcal{F}_{ij}| > \mathcal{F}_{\text{crit}}$) or remains hysteretic ($|\mathcal{F}_{ij}| < \mathcal{F}_{\text{crit}}$).

### 4.4. Margin-Inducing Training Objective

Our asymptotic analysis relies on a *force-separation* condition: compatible pairs should receive positive forces exceeding $\mathcal{F}_{\text{crit}}$ (up to a margin), while incompatible pairs should receive sufficiently negative forces. Under a sustained force-margin condition relative to $\mathcal{F}_{\text{crit}}$, the induced edge dynamics converge to a unique stable well, leading to topology polarization (Lemma A.4, Appendix A.6). To make this condition learnable, we introduce an explicit force-margin regularizer: let $\mathcal{L}_{task}$ denote the supervised objective (*e.g.*, cross-entropy for node classification). We construct positive and negative pair sets $\mathcal{P}$ and $\mathcal{N}$, respectively (with sampling for efficiency), and encourage a margin relative to the hysteresis threshold:

$$
\begin{aligned}
\mathcal{L}_{margin} = \sum_{(i,j)\in\mathcal{P}} \text{softplus}\Big(\big(\mathcal{F}_{\text{crit}}+\delta\big)-\mathcal{F}_{ij}\Big) \\
+ \sum_{(i,j)\in\mathcal{N}} \text{softplus}\Big(\big(\mathcal{F}_{\text{crit}}+\delta\big)+\mathcal{F}_{ij}\Big),
\end{aligned} \quad (13)
$$

where $\delta > 0$ is the margin hyperparameter. The total training objective is

$$\mathcal{L} = \mathcal{L}_{task} + \beta\,\mathcal{L}_{margin}, \quad (14)$$

with weight $\beta \geq 0$.

We use $\mathcal{L}_{margin}$ as an explicit mechanism to support the force-separation condition assumed in the theory. When labels are unavailable, we use a fixed pseudo-partition $\{c_i\}$ obtained by a lightweight clustering on initial features, and define $\mathcal{P} := \{(i,j) \in \mathcal{E}_{cand} : c_i = c_j\}$ and $\mathcal{N} := \{(i,j) \in \mathcal{E}_{cand} : c_i \neq c_j\}$. Note that this regularizer is designed to support the force-separation assumption; the task loss remains the primary supervision signal. Empirically, we find that task supervision alone may already push forces

toward separation, while $\mathcal{L}_{margin}$ improves stability and interpretability by aligning the learned force scale with the hysteresis threshold.

### 4.5. Static Candidate Set and Annealed Filtering

To scale HGODE, we evolve topology only on a fixed sparse candidate set $\mathcal{E}_{cand}$ constructed once at initialization (*e.g.*, 2-hop neighbors, Laplacian random-walk proposals, and random pairs). We maintain and integrate $\mathbf{U}_{ij}(t)$ only for $(i,j) \in \mathcal{E}_{cand}$, so the ODE state dimension is constant.

We use a monotone schedule $\mu(t)$ to progressively suppress weak candidate edges during training, yielding an implicit candidate set without discrete pruning. Operationally, $\mu(t)$ is applied inside the gating that forms the effective adjacency (Sec. 4.1), so edges that persist must be supported consistently by the learned force field.

## 5. Experiments

We conduct experiments to (i) validate the theoretical predictions of HGODE via controlled synthetic studies and (ii) demonstrate empirical effectiveness and robustness on real-world benchmarks. We first use Stochastic Block Model (SBM) graphs where ground-truth clusters are known, enabling direct inspection of infinite-depth behavior and topological phase separation. We then evaluate HGODE on standard node classification benchmarks and under graph perturbations.

**SBM generation.** We generate graphs from a $K$-block SBM with node partition $\{C_1, \ldots, C_K\}$ and block sizes $\{n_k\}_{k=1}^K$ ($\sum_k n_k = N$). For $i \in C_k$ and $j \in C_\ell$, edges are sampled independently as

$$\mathbf{A}_{ij}^{(0)} \sim \text{Bernoulli}(p_{k\ell}), \qquad p_{k\ell} = \begin{cases} p_{\text{in}}, & k = \ell, \\ p_{\text{out}}, & k \neq \ell, \end{cases}$$

with $p_{\text{in}} > p_{\text{out}}$ unless stated otherwise. Node features are initialized as

$$\mathbf{h}_i(0) = \mu_{c(i)} + \epsilon_i, \qquad \epsilon_i \sim \mathcal{N}(0, \sigma^2\mathbf{I}),$$

where $c(i) \in \{1, \ldots, K\}$ denotes the SBM block label.

**Models and baselines.** In synthetic experiments we compare (i) a soft-attention Graph ODE baseline and (ii) HGODE. The soft-attention baseline constructs a strictly positive, row-stochastic transition matrix

$$\mathbf{P}_{ij}(t) = \frac{\exp(s_{ij}(t)/\tau_{\text{attn}})}{\sum_{k=1}^N \exp(s_{ik}(t)/\tau_{\text{attn}})},$$

where $s_{ij}(t) = \langle \mathbf{h}_i(t), \mathbf{h}_j(t) \rangle$, and evolves features via consensus dynamics

$$\frac{d\mathbf{H}(t)}{dt} = -\big(\mathbf{I} - \mathbf{P}(t)\big)\mathbf{H}(t)$$

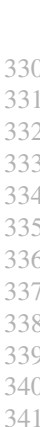
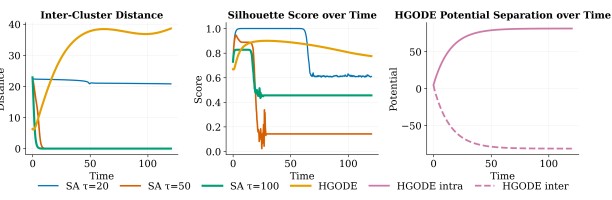
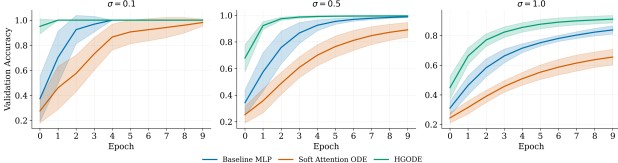

*Figure 2.* **Left**: Inter-cluster distance under soft attention collapses as $\tau_{\text{attn}}$ increases, indicating information leakage under increasingly diffuse mixing. **Middle**: The silhouette score decreases under soft attention at large $\tau_{\text{attn}}$, whereas HGODE remains stable at long time horizons. **Right**: HGODE potentials $\mathbf{U}_{ij}(t)$ polarize into positive (intra) and negative (inter) regimes, providing mechanistic evidence for hysteresis-driven topological separation.

HGODE follows Eq.7, with effective adjacency recovered by $\mathbf{A}_{ij}(t) = \sigma(\mathbf{U}_{ij}(t)/\tau)$.

## 5.1. Synthetic experiments

**Evaluation protocol and theory-driven diagnostics.** Our synthetic diagnostics are designed to directly probe the mechanisms predicted by the theory. To validate Theorem 3.3, we conduct an attention-temperature sweep over $\tau_{\text{attn}}$ for the soft-attention (SA) Graph ODE baseline and track the long-time evolution of node representations. We measure: (i) representation mixing via intra- and inter-cluster distances, (ii) clustering quality via the silhouette score over time, and (iii) for HGODE, the mean edge potential $\mathbf{U}_{ij}(t)$ for intra- and inter-cluster node pairs as a mechanistic readout. Figure 2 summarizes the results. As $\tau_{\text{attn}}$ increases, SA becomes increasingly diffuse and exhibits progressive information leakage, ultimately driving the features toward collapse. In contrast, HGODE preserves cluster distinguishability at long time horizons: inter-cluster separation remains large while intra-cluster dispersion is controlled. The potential trajectories further confirm hysteresis-induced phase separation, with $\mathbf{U}_{ij}(t)$ converging to a positive regime for intra-cluster pairs and to a negative regime for inter-cluster pairs, corresponding to connected versus insulated topological phases, respectively.

**Synthetic perturbation tests.** To evaluate robustness under feature corruption, we generate SBM graphs with increased cross-cluster connectivity by setting $p_{\text{out}} = 0.3$ (while keeping other SBM parameters fixed). We perturb node features by additive Gaussian noise with standard deviation $\sigma$, *i.e.*,

$$\mathbf{h}_i(0) = \mu_{c(i)} + \epsilon_i, \qquad \epsilon_i \sim \mathcal{N}(0, \sigma^2 \mathbf{I}),$$

and sweep $\sigma$ to control the signal-to-noise ratio. We generate 2000 graphs in total, using 80% for training and 20% for validation. We compare three models: a DeepSets-style MLP baseline (ignoring graph structure), the soft-attention Graph ODE baseline, and HGODE. For each model and each noise level, we run 10 random seeds and train for

*Figure 3.* Validation accuracy under feature perturbations on SBM graphs ($\mu = 0.5$, $p_{\text{out}} = 0.3$). Subplots correspond to noise levels **Left**: $\sigma = 0.1$, **Middle**: $\sigma = 0.5$, **Right**: $\sigma = 1.0$. Solid lines denote mean over 10 seeds and shaded regions indicate $\pm 1$ standard deviation.

10 epochs, reporting validation accuracy aggregated across seeds (mean with shaded uncertainty).

Figure 3 summarizes the results. As $\sigma$ increases, the soft-attention Graph ODE becomes markedly less robust: performance degrades and optimization becomes slower, consistent with increased representation mixing under diffuse global attention. In contrast, HGODE remains comparatively stable across noise levels, indicating that its dynamic topology can suppress spurious feature diffusion and recover informative signal even when the features are substantially corrupted (*e.g.*, at $\sigma = 1.0$ with $\mu = 0.5$, corresponding to a low signal-to-noise regime).

## 5.2. Real-world benchmark evaluation

We evaluate HGODE on node- and graph-level benchmarks where long-range propagation and long-time mixing behavior are most relevant. We defer link prediction to future work. For node classification task, we choose Cora (McCallum et al., 2000) and ogbn-proteins (Hu et al., 2020) for a general comparison, and Chameleon (Rozemberczki et al., 2021) as a heterophily condition validation; for graph classification task, we choose ZINC (Dwivedi et al., 2023), ogbg-molpcba (Hu et al., 2020), and additional Peptides-func (Dwivedi et al., 2022) as long-range graph dataset. All the experiments are conducted by NVIDIA A100 GPUs. The optimizer is Adam, and the learning rate varies from datasets for best performance.

**Baselines.** We compare against three families of methods: **(i) Message-passing GNNs:** GCN (Kipf & Welling, 2017), GCNII (Chen et al., 2020a), and GCN$^+$ (Luo et al., 2025); **(ii) Graph transformers:** GraphGPS (Rampášek et al., 2022), Polynormer (Deng et al., 2024), SGFormer (Wu et al., 2023a), and Subgraphormer (Bar-Shalom et al., 2024); **(iii) Continuous-depth models:** GDE (Poli et al., 2019), CGNN (Xhonneux et al., 2020), GRAND (Chamberlain et al., 2021), Sheaf Diffusion (Bodnar et al., 2022), and FLODE (Maskey et al., 2024). The last group is most closely aligned with our setting and directly tests whether HGODE improves long-time behavior over existing Graph ODE formulations.

*Table 1.* Performance evaluation on real-world benchmarks. – indicates not applicable / no official implementation for that task. ACC. refers to accuracy, and A.P. refers to average precision. We marked the top 3 performances as green, blue and grey color, respectively.

| | NODE CLASSIFICATION | | | GRAPH CLASSIFICATION | | |
|---|---|---|---|---|---|---|
| METRIC | CORA ACC.↑ | CHAMELEON ACC.↑ | OGBN-PROTEINS ROC-AUC↑ | ZINC MAE↓ | PEPTIDES-FUNC A.P.↑ | OGBG-MOLPCBA A.P.↑ |
| GCN | 81.42±0.36 | 40.27±2.48 | 71.18±0.27 | 0.359±0.015 | 0.683±0.004 | 0.204±0.002 |
| GCNII | 84.74±0.10 | 44.47±1.48 | 75.64±0.48 | 0.178±0.032 | **0.699±0.003** | 0.265±0.003 |
| GCN$^+$ | 85.27±0.55 | 45.89±2.86 | 77.84±0.56 | 0.087±0.012 | **0.716±0.005** | 0.269±0.002 |
| GRAPHGPS | 82.95±1.23 | 41.04±3.87 | 77.25±0.42 | **0.069±0.008** | 0.655±0.004 | **0.291±0.001** |
| POLYNORMER | 83.24±0.72 | 41.69±3.22 | 79.41±0.56 | – | – | – |
| SGFORMER | 84.59±0.43 | 45.77±3.48 | **79.84±0.61** | – | – | – |
| SUBGRAPHORMER | – | – | – | **0.082±0.04** | 0.661±0.005 | **0.293±0.004** |
| GDE | 82.44±0.81 | 45.63±2.47 | 74.63±0.20 | 0.239±0.012 | 0.632±0.014 | 0.247±0.002 |
| CGNN | 83.68±0.46 | 51.35±1.58 | 77.12±0.61 | 0.153±0.028 | 0.651±0.037 | 0.252±0.003 |
| GRAND | 83.61±0.37 | 57.72±1.86 | 76.45±0.42 | 0.147±0.018 | 0.663±0.016 | 0.260±0.003 |
| SHEAF DIFF. | **85.53±0.64** | **68.16±1.62** | 78.96±0.56 | 0.148±0.021 | 0.678±0.047 | 0.263±0.002 |
| FLODE | **86.44±1.17** | 67.26±1.37 | 79.23±0.75 | 0.124±0.034 | 0.674±0.041 | 0.267±0.004 |
| HGODE (OURS) | **86.26±0.78** | **72.56±1.24** | **81.24±0.63** | **0.078±0.025** | **0.714±0.022** | **0.278±0.003** |
| *w/o hysteresis* | 83.24±0.32 | 66.24±1.26 | 75.26±0.15 | 0.145±0.032 | 0.671±0.013 | 0.254±0.005 |
| *w/o topo. search* | 84.14±0.46 | **70.44±1.41** | 77.19±0.52 | 0.162±0.017 | 0.653±0.041 | 0.262±0.002 |
| *w/o force margin* | 84.36±0.19 | 61.24±0.73 | **80.24±0.77** | 0.172±0.080 | 0.689±0.034 | 0.260±0.003 |

**Solver choice.** The coupled dynamics can exhibit sharp switching near saddle-node transitions of the topology field (Sec. 3.4), where the vector field changes rapidly as $|F_{ij}|$ approaches the hysteresis threshold. We therefore use an adaptive-step Dormand–Prince solver (dopri5) for all experiments. The solver tolerance is set to both $1e^{-5}$.

**Main results and interpretation.** Table 1 summarizes the results under comparable parameter budgets ($\sim$1M for non-GCN models). HGODE consistently performs well across benchmarks and shows its largest advantages in regimes where dense, irreversible mixing is known to be harmful. In particular, HGODE achieves state-of-the-art performance on the heterophilous CHAMELEON dataset and obtains the best result on OGBN-PROTEINS, a long-range dependency benchmark. On standard homophilous datasets such as CORA, HGODE remains competitive with strong continuous-depth baselines, indicating that the proposed dynamics do not sacrifice performance in easier regimes. For graph-level tasks, HGODE is competitive with modern GNNs on long-range benchmarks, while remaining slightly behind highly specialized transformer-style models. Overall, these results support our central claim that coupling feature dynamics with hysteretic topology evolution is particularly beneficial for heterophily and long-time propagation.

**Ablation experiments.** We ablate three components to quantify their individual contributions: (i) **No-hysteresis**: we remove the cubic term in Eq. 4, turning the topology dynamics into a single-well relaxation; (ii) **No topology search**: we disable dynamic topology evolution and restrict propagation to the prior edges only; (iii) **No force margin**: we drop the force margin loss term from Eq. 14.

The results in Table 1 show that removing hysteresis consistently degrades performance on all datasets, supporting the role of bistability in preventing unstable edge flipping and in maintaining discriminative structure over long-time integration. Disabling topology search mainly harms performance on graphs requiring long-range dependency modeling, while the impact is smaller on relatively local graphs, suggesting that extended candidate connectivity is most useful when task-relevant signals are multi-hop. Finally, removing the force margin loss causes a substantial drop on CHAMELEON (15.6% absolute), indicating that the learned force field and its induced edge biases are critical under heterophily, where naive proximity-based neighborhoods are often misleading.

## 6. Conclusion

We identify the *monostability trap* in diffusion-style Graph ODEs, where strictly positive and irreducible mixing leads to inevitable feature collapse in the long-time regime. To address this issue, we propose **Hysteresis Graph ODE (HGODE)**, which treats graph topology as a co-evolving dynamical state rather than a fixed or softly weighted operator. By introducing latent edge potentials governed by bistable hysteresis, HGODE enables differentiable topological phase transitions that explicitly control the asymptotic mixing structure. This provides a principled mechanism to stabilize continuous-depth graph representation learning by dynamically shaping connectivity, instead of solely modifying feature propagation on a fixed graph. Future directions include exploring richer force parameterizations, more expressive latent potential landscapes, alternative candidate set constructions.

## Impact Statement

This paper presents work whose goal is to advance the field of Machine Learning. There are many potential societal consequences of our work, none which we feel must be specifically highlighted here.

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

## A. Proofs.

### A.1. Proof of Theorem 3.3.

*Proof.* The evolution of the node features is governed by the linear differential equation:

$$\frac{d\mathbf{H}(t)}{dt} = -\mathbf{L}_{rw}\mathbf{H}(t), \tag{15}$$

where $\mathbf{L}_{rw} = \mathbf{I} - \mathbf{P}$ is the random-walk Laplacian. Since $\mathbf{P}$ is time-invariant in this analysis, the analytical solution is given by the matrix exponential:

$$\mathbf{H}(t) = e^{-t(\mathbf{I}-\mathbf{P})}\mathbf{H}(0). \tag{16}$$

We analyze the asymptotic behavior of the operator $\mathbf{T}(t) = e^{-t(\mathbf{I}-\mathbf{P})}$ via the spectral properties of $\mathbf{P}$. Since $\mathbf{P}$ is row-stochastic and irreducible (by the strong connectivity assumption), the Perron-Frobenius theorem guarantees the following:

1. The spectral radius is $\rho(\mathbf{P}) = 1$, and $\lambda_1 = 1$ is a simple eigenvalue.

2. The right eigenvector associated with $\lambda_1$ is $\mathbf{1}$ (since $\mathbf{P1} = \mathbf{1}$).

3. The left eigenvector associated with $\lambda_1$ is the unique stationary distribution $\boldsymbol{\pi}$.

4. All other eigenvalues $\lambda_i$ (for $i = 2, \ldots, N$) satisfy $|\lambda_i| < 1$, implying $\text{Re}(\lambda_i) < 1$.

Let $\mathbf{P} = \mathbf{U}\boldsymbol{\Lambda}\mathbf{U}^{-1}$ be the eigendecomposition of $\mathbf{P}$ (assuming diagonalizability for exposition; the argument holds generally using Jordan canonical forms). The eigenvalues of the Laplacian $\mathbf{L}_{rw}$ are given by $\mu_i = 1 - \lambda_i$. Using the spectral expansion, we can express the evolution of the features as:

$$\mathbf{H}(t) = \left( \sum_{i=1}^{N} e^{-t(1-\lambda_i)} \mathbf{u}_i \mathbf{v}_i^{\top} \right) \mathbf{H}(0), \tag{17}$$

where $\mathbf{u}_i$ and $\mathbf{v}_i^{\top}$ are the corresponding right and left eigenvectors.

Separating the dominant mode ($i = 1$) from the non-dominant modes ($i \geq 2$):

$$\mathbf{H}(t) = e^{-t(1-1)}\mathbf{1}\boldsymbol{\pi}^{\top}\mathbf{H}(0) + \sum_{i=2}^{N} e^{-t(1-\lambda_i)} \mathbf{u}_i \mathbf{v}_i^{\top}\mathbf{H}(0). \tag{18}$$

For the dominant mode, the exponential term is $e^0 = 1$. For all non-dominant modes $i \geq 2$, since $\text{Re}(\lambda_i) < 1$, we have $\text{Re}(1 - \lambda_i) > 0$. Consequently, the exponential terms decay asymptotically:

$$\lim_{t\to\infty} \left| e^{-t(1-\lambda_i)} \right| = \lim_{t\to\infty} e^{-t\cdot\text{Re}(1-\lambda_i)} = 0. \tag{19}$$

Taking the limit $t \to \infty$ for the full system:

$$\lim_{t\to\infty} \mathbf{H}(t) = \mathbf{1}\boldsymbol{\pi}^{\top}\mathbf{H}(0) + \mathbf{0} = \mathbf{1}\left( \boldsymbol{\pi}^{\top}\mathbf{H}(0) \right). \tag{20}$$

Thus, the system converges to a state where every row of $\mathbf{H}$ is identical to the weighted centroid of the initial features, completing the proof. $\square$

### A.2. Proof of Lemma 3.5

We first prove a stronger differential contraction bound, from which the window contraction lemma follows immediately.

**Lemma A.1** (Instantaneous diameter contraction). *Let $x(t) \in \mathbb{R}^N$ follow the time-varying consensus flow*

$$\dot{x}(t) = -(I - P(t))x(t) = -x(t) + P(t)x(t),$$

*where $P(t)$ is row-stochastic for all $t$ and satisfies uniform positivity: $P_{ij}(t) \geq \alpha$ for all $i, j, t$ with some $\alpha > 0$. Define $\mathrm{diam}(x) := \max_i x_i - \min_i x_i$. Then for almost every $t$,*

$$\frac{d}{dt}\mathrm{diam}(x(t)) \leq -2\alpha\,\mathrm{diam}(x(t)).$$

*Consequently,*

$$\mathrm{diam}(x(t)) \leq e^{-2\alpha t}\,\mathrm{diam}(x(0)), \qquad \forall t \geq 0.$$

*Proof.* Let $M(t) := \max_i \mathbf{x}_i(t)$ and $m(t) := \min_i \mathbf{x}_i(t)$. Pick indices $i^+(t) \in \arg\max_i \mathbf{x}_i(t)$ and $i^-(t) \in \arg\min_i \mathbf{x}_i(t)$ (the argument holds for almost every $t$, since $M(t)$ and $m(t)$ are *a.e.* differentiable).

For the maximizer $i^+$, we have

$$\dot{\mathbf{x}}_{i^+}(t) = -\mathbf{x}_{i^+}(t) + \sum_{j=1}^N P_{i^+ j}(t)\,\mathbf{x}_j(t).$$

Let $j^-$ be any minimizer such that $\mathbf{x}_{j^-}(t) = m(t)$. By uniform positivity, $P_{i^+ j^-}(t) \geq \alpha$. Since all other $\mathbf{x}_j(t) \leq m(t)$, we obtain the upper bound

$$\sum_{j=1}^N P_{i^+ j}(t)\,\mathbf{x}_j(t) \leq (1-\alpha)m(t) + \alpha m(t).$$

Therefore,

$$\dot{M}(t) = \dot{\mathbf{x}}_{i^+}(t) \leq -M(t) + (1-\alpha)m(t) + \alpha m(t) = -\alpha\,(m(t) - m(t)).$$

Similarly, for the minimizer $i^-$, let $j^+$ be a maximizer with $\mathbf{x}_{j^+}(t) = m(t)$. Using $P_{i^- j^+}(t) \geq \alpha$ and $\mathbf{x}_j(t) \geq m(t)$ for all $j$, we obtain

$$\sum_{j=1}^N P_{i^- j}(t)\,\mathbf{x}_j(t) \geq (1-\alpha)m(t) + \alpha m(t),$$

hence

$$\dot{m}(t) = \dot{\mathbf{x}}_{i^-}(t) \geq -m(t) + (1-\alpha)m(t) + \alpha m(t) = \alpha\,(m(t) - m(t)).$$

Combining the two inequalities,

$$\frac{d}{dt}\mathrm{diam}(\mathbf{x}(t)) = \dot{M}(t) - \dot{m}(t) \leq -\alpha(M - m) - \alpha(M - m) = -2\alpha\,\mathrm{diam}(\mathbf{x}(t)).$$

Solving this differential inequality yields $\mathrm{diam}(\mathbf{x}(t)) \leq e^{-2\alpha t}\mathrm{diam}(\mathbf{x}(0))$. $\square$

**Lemma A.2** (Window contraction). *Under the assumptions of Lemma A.1, for any $T_w > 0$ there exists $\rho \in (0,1)$ such that*

$$\mathrm{diam}(x(t + T_w)) \leq (1 - \rho)\,\mathrm{diam}(x(t)), \qquad \forall t \geq 0.$$

*One can take $\rho := 1 - e^{-2\alpha T_w}$.*

*Proof.* By Lemma A.1, $\mathrm{diam}(\mathbf{x}(t + T_w)) \leq e^{-2\alpha T_w}\mathrm{diam}(\mathbf{x}(t))$. Set $\rho = 1 - e^{-2\alpha T_w} \in (0,1)$. $\square$

### A.3. Proof of Theorem 3.6

**Theorem A.3** (Consensus trap under time-varying positive mixing)**.** *Consider the time-varying consensus flow*

$$\frac{d\boldsymbol{H}(t)}{dt} = -(\boldsymbol{I} - \boldsymbol{P}(t))\boldsymbol{H}(t),$$

*where $\boldsymbol{P}(t)$ is row-stochastic and uniformly positive: $\boldsymbol{P}_{ij}(t) \geq \alpha$ for all $i, j, t$. Then for each feature dimension $k$, there exist constants $C, \kappa > 0$ such that*

$$\mathrm{diam}(\boldsymbol{H}_{\cdot k}(t)) \leq Ce^{-\kappa t}\,\mathrm{diam}(\boldsymbol{H}_{\cdot k}(0)).$$

*Moreover, there exists $y(t) \in \mathbb{R}^m$ such that*

$$\lim_{t \to \infty} \|\boldsymbol{H}(t) - \mathbf{1}y(t)^\top\|_F = 0.$$

*Proof.* Fix a feature dimension $k \in \{1, \ldots, m\}$ and define $\mathbf{x}(t) := \boldsymbol{H}_{\cdot k}(t) \in \mathbb{R}^N$. From the matrix ODE, $x(t)$ satisfies

$$\dot{\mathbf{x}}(t) = -(\mathbf{I} - \mathbf{P}(t))\mathbf{x}(t).$$

Applying Lemma A.1 yields

$$\mathrm{diam}(\mathbf{H}_{\cdot k}(t)) = \mathrm{diam}(\mathbf{x}(t)) \leq e^{-2\alpha t}\,\mathrm{diam}(\mathbf{x}(0)) = e^{-2\alpha t}\,\mathrm{diam}(\mathbf{H}_{\cdot k}(0)).$$

Thus we can take $C = 1$ and $\kappa = 2\alpha$.

To prove convergence to the rank-one consensus subspace, define $y(t) \in \mathbb{R}^m$ as the first row of $\mathbf{H}(t)$, i.e., $y(t)^\top := \mathbf{H}_{1\cdot}(t)$. Then for any node $i$ and any feature dimension $k$,

$$|\mathbf{H}_{ik}(t) - y_k(t)| \leq \mathrm{diam}(\mathbf{H}_{\cdot k}(t)).$$

Therefore,

$$\|\mathbf{H}(t) - \mathbf{1}y(t)^\top\|_F^2 = \sum_{i=1}^{N}\sum_{k=1}^{m}|\mathbf{H}_{ik}(t) - y_k(t)|^2 \leq \sum_{i=1}^{N}\sum_{k=1}^{m}\mathrm{diam}(\mathbf{H}_{\cdot k}(t))^2 = N\sum_{k=1}^{m}\mathrm{diam}(\mathbf{H}_{\cdot k}(t))^2.$$

Since each $\mathrm{diam}(\mathbf{H}_{\cdot k}(t))$ decays exponentially to 0, the right-hand side vanishes as $t \to \infty$, implying $\|\mathbf{H}(t) - \mathbf{1}y(t)^\top\|_F \to 0$. $\square$

### A.4. Proof of Corollary 3.7

*Proof.* Recall the definition of the entries of the row-stochastic transition matrix $\mathbf{P}(t)$ for soft attention:

$$\mathbf{P}_{ij}(t) = \frac{\exp(\langle \mathbf{h}_i(t), \mathbf{h}_j(t)\rangle/\tau)}{\sum_{k=1}^{N}\exp(\langle \mathbf{h}_i(t), \mathbf{h}_k(t)\rangle/\tau)}. \tag{21}$$

By the Cauchy-Schwarz inequality and the boundedness assumption $\|\mathbf{h}_i(t)\| \leq B$, the inner product is bounded by:

$$|\langle \mathbf{h}_i(t), \mathbf{h}_j(t)\rangle| \leq \|\mathbf{h}_i(t)\|\|\mathbf{h}_j(t)\| \leq B^2. \tag{22}$$

Consequently, for any pair $(i, j)$, the unnormalized attention score is bounded within:

$$\exp(-B^2/\tau) \leq \exp(\langle \mathbf{h}_i(t), \mathbf{h}_j(t)\rangle/\tau) \leq \exp(B^2/\tau). \tag{23}$$

Now we bound the denominator (the partition function $Z_i$). Since the sum contains $N$ terms, and each term is bounded from above by $\exp(B^2/\tau)$:

$$\sum_{k=1}^{N}\exp(\langle \mathbf{h}_i(t), \mathbf{h}_k(t)\rangle/\tau) \leq N\exp(B^2/\tau). \tag{24}$$

Substituting the lower bound of the numerator and the upper bound of the denominator into the expression for $\mathbf{P}_{ij}(t)$:

$$\mathbf{P}_{ij}(t) \geq \frac{\exp(-B^2/\tau)}{N\exp(B^2/\tau)} = \frac{1}{N}\exp\left(-\frac{2B^2}{\tau}\right). \tag{25}$$

Let $\alpha = \frac{1}{N}\exp(-\frac{2B^2}{\tau})$. Since $B$ is finite and $\tau > 0$, we have $\alpha > 0$. Thus, $\mathbf{P}_{ij}(t) \geq \alpha$ for all $i, j, t$, satisfying Assumption 3.4 (Uniform Positivity). By Theorem 3.6, the dynamics inevitably converge to the rank-one consensus subspace. $\square$

### A.5. Proof of Proposition 3.9

*Proof.* Equilibria satisfy $u_{ij} - u_{ij}^3 + \mathcal{F}_{ij} = 0$, *i.e.*, (5). At a saddle-node transition, the cubic has a double root, so both $f(u) := u^3 - u - \mathcal{F}_{ij} = 0$ and $f'(u) = 3u^2 - 1 = 0$ hold. Thus $u = \pm 1/\sqrt{3}$ and $\mathcal{F}_{ij} = u^3 - u = \pm 2/(3\sqrt{3})$, giving (6). The stated stability and regime characterization follow from standard one-dimensional bifurcation analysis of the forced double-well potential. $\square$

### A.6. Force margin implies polarization of edge potentials

**Lemma A.4** (Force margin implies global polarization). *Consider the scalar edge-potential dynamics*

$$\dot{u} = u - u^3 + F, \tag{26}$$

*where $F \in \mathbb{R}$ is constant and $F_{\mathrm{crit}} := \frac{2}{3\sqrt{3}}$. If $|F| > F_{\mathrm{crit}}$, then (26) admits a* unique *equilibrium $u^\star(F)$, and this equilibrium is* globally asymptotically stable. *Moreover, $\mathrm{sign}(u^\star(F)) = \mathrm{sign}(F)$. In particular, if $|F| \geq F_{\mathrm{crit}} + \delta$ for some $\delta > 0$, then every trajectory satisfies $u(t) \to u^\star(F)$ as $t \to \infty$, i.e., the potential polarizes to the connected well for $F > 0$ and to the insulated well for $F < 0$.*

*Proof.* Define $g(u) := u - u^3 + F$. Equilibria are roots of $g(u) = 0$, i.e., of the cubic $u^3 - u = F$. By Proposition 3.9, when $|F| > F_{\mathrm{crit}}$ the system is in the monostable regime, so the cubic has exactly one real root; denote it by $u^\star(F)$.

Local stability follows from the linearization $g'(u) = 1 - 3u^2$. In the monostable regime, the unique equilibrium must be stable, hence $g'(u^\star) < 0$.

To see global convergence, note that $g(u)$ is continuous and has a unique zero at $u^\star$. Therefore, $g(u)$ has a fixed sign on each side of $u^\star$: $g(u) < 0$ for all $u > u^\star$ and $g(u) > 0$ for all $u < u^\star$ (otherwise an additional root would exist). Hence, if $u(0) > u^\star$, then $\dot{u} = g(u) < 0$ until the trajectory approaches $u^\star$; similarly, if $u(0) < u^\star$, then $\dot{u} > 0$. Thus $u(t)$ is monotone and bounded, so it converges, and the only possible limit is the equilibrium $u^\star$. This proves global asymptotic stability.

Finally, $\mathrm{sign}(u^\star(F)) = \mathrm{sign}(F)$ follows from $g(0) = F$ together with $\lim_{u \to +\infty} g(u) = -\infty$ and $\lim_{u \to -\infty} g(u) = +\infty$: for $F > 0$ the unique root lies in $(0, \infty)$, and for $F < 0$ it lies in $(-\infty, 0)$. $\square$