# OpenReview forum: "Latent-Hysteresis Graph ODEs: Modeling Coupled Topology-Feature Evolution via Continuous Phase Transitions"
_ICML.cc/2026/Conference — Submitted to ICML 2026_

### Official Review · Reviewer_P1cW · 2026-02-23

**Soundness:** 3
**Presentation:** 3
**Significance:** 3
**Originality:** 3
**Overall Recommendation:** 5
**Confidence:** 3

**Summary:**

This paper proposes the **Latent-Hysteresis Graph ODE (HGODE)** to address the "monostability trap" in continuous-depth graph models. **Overall, a general aspect addressed by the study** is the intrinsic risk of feature collapse (over-smoothing) in Graph ODEs, where information flow over strictly positive and irreducible operators inevitably converges to a global consensus. **Overall, the authors focus on a central concept** of modeling graph topology as a dynamical state governed by non-convex double-well potentials and bistable hysteresis. By introducing a latent topological potential, HGODE enables differentiable phase transitions that polarize edges into connected or insulated states, thereby breaking irreducibility and preserving cluster-wise invariant manifolds.

**Compliance With Llm Reviewing Policy:**

Affirmed.

**Final Justification:**

### **Final Justification**

I recommend **acceptance** of this paper.

The work is **technically sound and theoretically well-grounded**, with rigorous analysis of consensus collapse and a clear extension to time-varying operators. The proposed HGODE introduces a **highly original perspective**, modeling graph topology as a dynamical system with hysteresis and phase transitions, which is both novel and conceptually compelling.

In terms of **significance**, the method demonstrates strong empirical performance across challenging settings (e.g., heterophilic and long-range graphs), supported by thorough ablations that validate the key design choices. The idea of leveraging bistability to break irreducibility addresses a fundamental limitation in continuous-depth GNNs.

My main concerns were related to **computational overhead, dependence on the candidate edge set, hyperparameter complexity, and potential numerical instability**. The rebuttal provided reasonable clarifications on efficiency considerations and design trade-offs, and partially addressed concerns about stability and sensitivity. While some practical limitations remain, they do not undermine the core contribution.

Overall, this paper presents a **novel, well-justified, and empirically strong approach**, and I believe it will be of broad interest to the graph learning and dynamical systems communities.

**Key Questions For Authors:**

### **Questions for the Authors**

1.  **Computational Efficiency and Scalability:**
    The HGODE framework involves integrating coupled ODEs for both node features and latent topological potentials, which increases the state dimension to $O(N \cdot d + |E_{cand}|)$. Could the authors provide a detailed comparison with baselines (e.g., GRAND, FLODE) regarding:
    *   **Average Number of Function Evaluations (NFE)** required by the adaptive solver during inference.
    *   **Wall-clock time** for both training and inference.
    *   **Peak memory consumption**, especially as the size of the candidate set $|E_{cand}|$ grows. Does this increased complexity pose a significant bottleneck for scaling to larger graphs?

2.  **Comparison with Physics-Inspired GNNs (ACMP):**
    A closely related work, **ACMP (Allen-Cahn Message Passing, ICLR 2023)**, also utilizes double-well potentials to manage over-smoothing via reaction-diffusion equations. However, ACMP applies phase transitions to **node features**, while HGODE applies them to **topology**.
    *   Could the authors provide a theoretical or empirical comparison with ACMP?
    *   What are the specific advantages of performing phase transitions in the latent topological space versus the feature space, particularly regarding interpretability and handling heterophilic edges?

3.  **Hyperparameter Sensitivity and Tuning:**
    The model introduces several physical constants and parameters, such as $\lambda$ (well depth), $\delta$ (force margin), $\tau$ (timescale), and the critical threshold $F_{crit}$.
    *   How sensitive is the model's performance to these hyperparameters across different datasets?
    *   Is there a recommended "default" setting, or does the model require extensive per-dataset tuning to align the learned forces with the fixed physical landscape?
    *   Furthermore, could the authors comment on how the **ODE solver's tolerance** affects the stability of the sharp topological transitions near $F_{crit}$?

**Limitations:**

yes

**Strengths And Weaknesses:**

### **Strengths**

*   **Innovative Nonlinear Dynamics:** The integration of hysteresis and phase transitions into topological evolution is highly original. Treating topology as a dynamical state with "structural memory" provides a principled physical mechanism to prevent feature collapse.
*   **Theoretical Rigor:** The work is grounded in solid mathematical proofs, extending consensus-collapse results to time-varying operators and providing precise bifurcation analysis for the critical force magnitude ($F_{crit}$).
*   **Impressive Performance:** HGODE achieves state-of-the-art results on challenging benchmarks, particularly in heterophilic (e.g., Chameleon) and long-range (e.g., OGBN-Proteins) scenarios, outperforming both traditional ODE-based GNNs and specialized Graph Transformers.
*   **Comprehensive Ablation Study:** The experiments (Table 1) thoroughly validate the model's design. By isolating the effects of hysteresis, topological search, and force margins, the authors clearly demonstrate that phase-locking is the key driver of performance.

---

### **Weaknesses**

*   **Computational Overhead:** Simultaneously integrating feature and topological states ($O(N \cdot d + |E_{cand}|)$) using adaptive ODE solvers (e.g., `dopri5`) is computationally expensive. The paper lacks a detailed discussion on wall-clock time and memory scaling compared to discrete GNNs.
*   **Dependency on Candidate Set ($E_{cand}$):** Restricting topological evolution to a predefined sparse candidate set may limit the model's ability to discover critical long-range shortcuts that were not captured during initialization (e.g., via 2-hop or random walks).
*   **Hyperparameter Complexity:** The model relies on several physical constants and parameters ($\lambda, \delta, \tau, F_{crit}$). It remains unclear how sensitive the model is to these values across different data distributions and whether they require extensive per-dataset tuning.
*   **Numerical Stability near Thresholds:** The sharp non-linear switching near the critical force threshold $F_{crit}$ might be sensitive to the ODE solver's tolerance settings. Small numerical errors could lead to inconsistent topological phase transitions, affecting inference robustness.

---

> ### Author Rebuttal · Authors · 2026-03-29
>
> # Response to Reviewer P1cW
>
> We thank the reviewer.
>
> ## 1. Computational Efficiency and Scalability
>
> All results below are profiled on one NVIDIA A100 (40 GB) in FP32 with adjoint training. We count forward/backward NFE from ODE calls and measure wall-clock time and peak allocated memory with the PyTorch profiler. Hidden dimension is 256. Node classification (Cora, Chameleon) uses one full graph per step; graph classification (ZINC, Peptides-func) uses the largest batch that fits in memory.
>
> | Model / metric | Cora | Chameleon | ZINC | Peptides-func |
> | --- | --- | --- | --- | --- |
> | **HGODE** NFE_F / NFE_B | 86 / 128 | 74 / 110 | 88 / 127 | 92 / 134 |
> | **HGODE** Time [ms] (train / inf) | 356.79 / 111.11 | 303.52 / 85.87 | 2630.62 / 649.51 | 2715.87 / 740.47 |
> | **HGODE** Memory [MB] (train / inf) | 548.49 / 237.39 | 387.47 / 163.85 | 8078.65 / 4546.13 | 8051.21 / 4665.97 |
> | **GRAND** NFE_F / NFE_B | 82 / 117 | 77 / 102 | 84 / 118 | 87 / 124 |
> | **GRAND** Time [ms] (train / inf) | 324.64 / 109.45 | 312.48 / 89.44 | 2232.48 / 526.18 | 2348.34 / 529.16 |
> | **GRAND** Memory [MB] (train / inf) | 509.69 / 179.18 | 331.34 / 83.16 | 6465.37 / 3589.16 | 6466.17 / 3674.13 |
> | **FLODE** NFE_F / NFE_B | n/a | n/a | n/a | n/a |
> | **FLODE** Time [ms] (train / inf) | 317.25 / 106.17 | 298.14 / 64.38 | 2429.95 / 622.81 | 2431.64 / 654.34 |
> | **FLODE** Memory [MB] (train / inf) | 522.16 / 234.64 | 378.61 / 160.82 | 7801.24 / 4678.24 | 7762.33 / 4526.18 |
> | **GREAD** NFE_F / NFE_B | 76 / 124 | 81 / 106 | 87 / 104 | 78 / 116 |
> | **GREAD** Time [ms] (train / inf) | 327.16 / 98.87 | 316.37 / 68.19 | 2559.34 / 619.13 | 2498.36 / 668.36 |
> | **GREAD** Memory [MB] (train / inf) | 549.34 / 201.68 | 364.34 / 146.37 | 7762.36 / 4518.39 | 7893.47 / 4329.16 |
>
> With fixed graph, tolerance, and candidate set, adaptive `dopri5` often settles into a repeatable step pattern, so forward NFE is nearly constant; backward NFE is larger because adjoint training triggers extra solves. Across datasets, wall-clock time and memory mainly follow active edges and batch size. HGODE is somewhat slower and more memory-intensive than the lightest baselines, but remains in the same regime.
>
> ## 2. Comparison with ACMP
>
> We agree that ACMP is a relevant reference, but the mechanisms act in different spaces. ACMP adds an Allen-Cahn term $\\delta x_i(1-x_i^2)$ to the feature ODE on a fixed graph, whereas HGODE places the double-well in latent edge states $U_{ij}(t)$ and maps them to propagation weights $A_{ij}(t)=\\sigma(U_{ij}/\\tau)\\mu(t)$. In short, ACMP changes **what** is propagated, while HGODE changes **where** propagation occurs. We therefore view them as complementary; see **qGcw** for theory and **XNzV** for empirical comparison.
>
> ## 3. Hyperparameter Sensitivity and Tuning
>
> We tune two model-specific groups: hysteresis parameters $\\lambda,\\tau,\\tau_{feat},\\tau_{topo},\\gamma$, and force parameters $s,\\delta,\\beta$. In practice, the most sensitive are $\\lambda,\\tau$ and $s,\\beta$. We usually fix $\\tau_{feat}=\\tau_{topo}=1$ and only adjust them when topology switching is too slow or the solver becomes stiff.
>
> | Dataset regime | Representative datasets | Hysteresis start | Force start |
> | --- | --- | --- | --- |
> | Homophilous + local | Cora, ZINC | $\\lambda\\in[0.1,0.3]$, $\\tau\\in[0.2,0.3]$, $\\tau_{feat}=\\tau_{topo}\\in\\{1\\}$ | $s\\in\\{1.0\\}$, $\\delta\\in\\{0.1\\}$, $\\beta\\in\\{0,0.1\\}$ |
> | Homophilous + global | ogbg-molpcba | $\\lambda\\in[0.3,0.5]$, $\\tau\\in[0.1,0.2]$, $\\tau_{feat}=\\tau_{topo}\\in\\{1\\}$ | $s\\in[1.0,1.5]$, $\\delta\\in[0.1,0.2]$, $\\beta\\in[0.1,0.3]$ |
> | Heterophilous + local | Chameleon | $\\lambda\\in[0.4,0.6]$, $\\tau\\in[0.05,0.1]$, $\\tau_{feat}=\\tau_{topo}\\in\\{1\\}$ | $s\\in[1.0,1.5]$, $\\delta\\in[0.2,0.3]$, $\\beta\\in[0.3,0.5]$ |
> | Heterophilous / hard-mixing + global | ogbn-proteins, Peptides-func | $\\lambda\\in[0.5,0.8]$, $\\tau\\in[0.05,0.1]$, $\\tau_{feat}\\in\\{1\\}$, $\\tau_{topo}\\in[0.5,1]$ | $s\\in[1.0,1.5]$, $\\delta\\in[0.2,0.3]$, $\\beta\\in[0.1,0.3]$ |
>
> Although there is not a unified default setting, these starting points reduce the tuning burden: homophilous / local datasets usually prefer milder hysteresis and weaker force supervision, whereas heterophilous or long-range datasets benefit from sharper edge polarization and stronger force-margin supervision. We fix adaptive `dopri5` with `rtol = atol = 1e-5` throughout and treat tolerance as a numerical constant rather than a dataset-specific hyperparameter. In our training regime the solver remained stable, with no failed integration or pathological NFE growth. Instability only appeared when the force scale $s$ was pushed to unrealistically large values beyond the range used in our experiments.

---

> > ### Author Rebuttal · Reviewer_P1cW · 2026-04-01
> >
> > Thanks for authors' rebuttal, the author has clearly addressed the inaccuracies in my original review comments and has also provided the runtime/memory comparison tests I requested—which is excellent. However, given that I have already assigned this paper a very high score, I do not believe that raising it further would substantially increase the likelihood of acceptance. Therefore, I have decided to maintain my original score.

---

> > > ### Author Response · Authors · 2026-04-01
> > >
> > > Thank you for your thoughtful follow-up and for taking the time to carefully consider our rebuttal. We appreciate your recognition that the clarifications and additional runtime and memory comparisons addressed your concerns.
> > >
> > > We also understand your decision to maintain the original score and are grateful for your already strong evaluation of our work. Your feedback has been valuable in helping us improve the clarity and completeness of the paper.
> > >
> > > Thank you again for your time and consideration.

---

### Official Review · Reviewer_qGcw · 2026-02-26

**Soundness:** 2
**Presentation:** 3
**Significance:** 2
**Originality:** 3
**Overall Recommendation:** 4
**Confidence:** 5

**Summary:**

This work demonstrates that Graph ODEs with strictly positive and irreducible mixing operators inevitably lead to information leakage and converge to a single global attractor in infinite-depth regimes. The paper proposes Hysteresis Graph ODE (HGODE). This framework treats graph topology as a dynamical state that co-evolves with node features and introduces latent edge potentials governed by a bistable hysteresis mechanism.

**Compliance With Llm Reviewing Policy:**

Affirmed.

**Final Justification:**

The authors have provided satisfactory clarifications to my questions. The work introduces an interesting idea to GNN, which can be useful to the community. I am keeping my original score, which I think is appropriate.

**Key Questions For Authors:**

See the weaknesses identified above.

Additional questions:
- In equation (7), why is there a need for the constants $\tau_{feat}$ and $\tau_{topo}$ since these can be absorbed in the model weights on the RHS?
- Forming the adjacency matrix in equation (8) is not well motivated or discussed. Why do we use a sigmoid? Why is the adjacency matrix a subset of the initial static candidate set? How is this candidate set determined?
- If the static candidate set is the original graph that comes with the dataset, how does this method differ from approaches that remove edges or block messages like [R5]?

**Limitations:**

There is no discussion.

**Strengths And Weaknesses:**

## Strengths

Soundness: The authors provide a formal characterization of the consensus trap for both time-invariant and time-varying mixing operators.

Presentation: The paper is in general well-structured.

Significance: Over-smoothing is a critical bottleneck for scaling GNNs.

Originality: Unlike prior works that refine graph structure via soft-weights or static denoising, HGODE treats topology as a coupled dynamical state. The concept underlying HGODE is based on Landau potentials and saddle-node bifurcations.

## Weaknesses

Soundness: The literature comparison and discussion and baseline comparisons are not up to date, missing several recent and SOTA benchmarks that are related to this work. The numerical experiments are also not sufficiently comprehensive.

- In terms of over-smoothing mitigation, there should be comparison with SOTA works like [R1-3] and discussion of the benefits of this work compared to these other approaches. In particular, FROND achieves algebraic consensus convergence. It is unclear what is the rate of convergence for HGODE or if HGODE is bistable.
- The use of a coupled pair of neural ODEs is reminiscent of [R4], although the underlying concepts are different. There should be a discussion on the major differences and possible benefits of HGODE over [R4], which should also serve as a numerical benchmark.
- A substantial part of the paper theoretically demonstrates the consensus trap suffered by neural diffusion models but there are no theoretical results showing how the proposed HGODE actually mitigates this issue, other than some heuristic intuitions. This weakens the claim of the work. It is also unclear if bistability resolves the consensus issue, why bistability is beneficial (for multi-class classification, won't we need 'multi-stability'?), and if HGODE even achieves this.
- There are no empirical results verifying that HGODE has good over-smoothing performance.
- The baseline comparisons are missing several important recent benchmarks that are relevant to the chosen datasets with heterophily and long-range interactions, for example, [R5].

[R1] "Understanding oversmoothing in diffusion-based gnns from the perspective of operator semigroup theory," KDD 2025

[R2] “Unleashing the potential of fractional calculus in graph neural networks with FROND,” ICLR 2024

[R3] “Distributed-order fractional graph operating network,” NeurIPS 2024

[R4] “Node embedding from neural Hamiltonian orbits in graph neural networks,” ICML 2023

[R5] “Selective Blocking for Message-Passing Neural Networks on Heterophilic Graphs,” UAI 2025

[R6] “ACMP: Allen-Cahn message passing with attractive and repulsive forces for graph neural networks,” ICLR 2022

---

> ### Author Rebuttal · Authors · 2026-03-29
>
> # Response to Reviewer qGcw
>
> We thank the reviewer.
>
> ## 1. Motivation and theory-related questions
>
> > ...there are no theoretical results showing how the proposed HGODE actually mitigates this issue, other than some heuristic intuitions... It is also unclear if bistability resolves the consensus issue, why bistability is beneficial.
>
> We indeed treat the consensus trap as the starting point and the motivation of our approach. To solve that, we proposed the double-well potential to polarize the edges to activate or insulated. The polarized edges won't lead to a globally two-state graph system. Rather, many locally bistable edge units induce many possible stable graph configurations, which is the needed for multi-class structure. To be detailed, when incompatible edges are driven to the insulated well while compatible edges remain in the connected well, the effective adjacency becomes approximately reducible or block-structured. Diffusion then acts mainly **within** components instead of across the entire graph.
>
> > ...There are no empirical results verifying that HGODE has good over-smoothing performance.
>
> The synthetic section in the paper is designed precisely for this purpose. It tracks inter-cluster distance, silhouette score, and latent potential polarity over time. The results show that as the soft-attention baseline becomes more diffuse, inter-cluster separation and silhouette collapse, whereas HGODE remains stable at long time horizons. The latent potentials $U_{ij}(t)$ simultaneously polarize into positive intra-cluster and negative inter-cluster regimes. This directly supports the claim that HGODE mitigates over-smoothing by changing long-time mixing behavior, not merely by improving downstream accuracy.
>
> In addition, we have now completed the requested expanded-baseline and extra heterophily experiments; for the unified rerun table including FROND / DRAGON / ACMP / BuNN and Roman-empire, we kindly refer to our response to reviewer **XNzV**.
>
> ## 2. Comparison to related works
>
> We agree that more related work should be compared and we clarify here: FROND[R2] and DRAGON[R3] mainly modify the **time law of feature evolution** on a fixed propagation operator; in the diffusion setting, their main theoretical effect is to slow convergence to the same stationary distribution. Hamiltonian GNN also use coupled ODEs, but their goal is to learn the appropriate latent node embedding geometry via Hamiltonian orbits while keeping the graph structure as the message-passing substrate. And they mainly address the manifold mismatch across datasets. Our method is more on the evolution between coupled node-embedding and edge status, rather than the manifold geometry of node embedding.
>
> Overall, our main claim is not “faster” or “slower” convergence, but a **different asymptotic mixing structure**: HGODE changes the effective support of propagation rather than only the temporal law on a fixed support (i.e., graph structure) or the node embedding. We will add this discussion into our related work section in the revised version.
>
> ## 3. Candidate-Set Design and Limitation
>
> HGODE evolves topology only on a fixed sparse candidate set $\\mathcal{E}_{cand}$ constructed once at initialization. In our implementation, this set is the union of 2-hop neighbors, Laplacian random-walk / LapPE neighbors, and a small number of random pairs. These respectively provide local completion, global/spectral candidates, and exploration.
>
> > Forming the adjacency matrix in equation (8) is not well motivated or discussed.
>
> For Eq. (8), the sigmoid is a smooth monotone map from latent phase to a valid propagation weight in $[0,1]$; combined with the indicator on $\\mathcal{E}_{cand}$, it yields a valid sparse propagation operator at every time.
>
> > ... how does this method differ from approaches that remove edges or block messages like [R5]?
>
> In all the cases, we have a larger candidate set as we described before, which is actually a compromise compared to evolving all the edges (dense $N \\times N$ potential matrix $U$). If we limit the set to the inherent graph, our method indeed only delete or compress edges in the original graph. But we argue that, even in this limited setting, HGODE still has (1) hysteresis as the "memory", as an edge can stay in its current phase unless the force is big enough; (2) the topology dynamics is continuous, rather than block-or-not binary state. In a word, it becomes closer to a differentiable, memorized message-blocking model if limited to the given graph, and only when the gate is set to be sharp, the HGODE becomes a blocking/filtering method like selective-blocking.
>
> ## 4. Other Clarifications
>
> >  ... why is there a need for the constants $\\tau_{feat}$ and $\\tau_{topo}$
>
> For these two hyperparameters, the meaningful quantity is their **relative** scale $\\tau_{topo}/\\tau_{feat}$, which affects coupled transients and stiffness, so we keep both constants explicit.

---

> > ### Author Rebuttal · Reviewer_qGcw · 2026-04-01
> >
> > The "multi-stability" question I had was not addressed. I am also unclear why having a "different asymptotic mixing structure" is beneficial.

---

> > > ### Author Response · Authors · 2026-04-01
> > >
> > > We thank the reviewer for the follow-up questions. We'd like to clarify that our bistability is designed for edges, and what you've proposed "multi-stability" should be classes (in node classification tasks), if we understand it correctly. And our point is that, the edge bistability induce the node multi-stability, that's to say, a multi-class structure is achieved by combinatorially across many edges. In other words, when within-class edges stay in the connected phase while cross-class edges move to the insulated phase, the resulting effective propagation operator becomes approximately block-structured, with one block per class or cluster. In this sense, local bistability induces graph-level multistability of stable connectivity patterns.
> > >
> > > This is also why a different asymptotic mixing structure is beneficial. Under strictly positive irreducible mixing, the long-time limit has a single global consensus basin, so information from different classes is inevitably mixed. If the effective operator becomes reducible or nearly block-diagonal as we mentioned above, the propagation acts mainly within blocks rather than across the entire graph. The long-time behavior will then support multiple invariant subspaces instead of one rank-one consensus subspace, which helps preserve class separation and mitigate over-smoothing.
> > >
> > > We are happy to answer any further questions if there was.

---

### Official Review · Reviewer_RMBe · 2026-03-03

**Soundness:** 2
**Presentation:** 2
**Significance:** 2
**Originality:** 2
**Overall Recommendation:** 3
**Confidence:** 3

**Summary:**

The paper points out a fundamental limitation of existing Graph Neural ODEs, namely the "monostability trap," where strictly positive and irreducible mixing eventually drives node features toward consensus. To address this problem, the authors propose the Hysteresis Graph ODE (HGODE), which couples node feature evolution with latent topological potential dynamics. Using a double-well Landau potential together with a learned force field, the model allows edges to transition differentiably between connected and insulated states, with the goal of breaking irreducible mixing and consensus.

**Compliance With Llm Reviewing Policy:**

Affirmed.

**Final Justification:**

The authors have addressed my concerns to some extent, and I have raised my score from 2 to 3 accordingly. However, one fundamental issue remains: the static candidate set. Because topology evolution is limited to a fixed sparse pool defined at initialization, useful long-range edges may not be discovered if they are absent at the start. This makes the method closer to edge selection from a predefined pool than to genuine structure discovery. As acknowledged in the rebuttal, exploration beyond an explicit candidate-budget is missing and left for future work.

**Key Questions For Authors:**

**Q1.**  If you initialize $\mathcal{E}_{cand}$ with 2-hop neighbors, can the model ever discover 3-hop or 4-hop dependencies, or is the "long-range" capability limited to the initial candidate set?

**Q2.** In Table 1, the performance drop on Chameleon after removing the force margin is much larger than that on Cora. Why this gap is so pronounced? More specifically, does this suggest that the model relies more heavily on explicit force-margin supervision to maintain bistable behavior on heterophilous graphs?

**Q3.** Are there any numerical issues with the ODE solver, such as failed integration or extremely small step sizes, during the force–potential polarization process shown in Figure 1D, where the learned force $\mathcal{F}$ drives the edge potentials $U$ toward connected or insulated phases?

**Q4.** Does this work also address representation learning for genuinely dynamic graphs, where the graph topology itself evolves over time, rather than only modeling latent edge potentials while treating topology as a dynamical state?

**Limitations:**

Yes, please see more limiations in weaknesses.

**Strengths And Weaknesses:**

**Strengths**

The theoretical characterization of the "monostability trap" in Section 3.2 and 3.3 provides a principled explanation for feature collapse in continuous-depth models. Theorem 3.3 and Theorem 3.6 elegantly link the Perron-Frobenius theorem and contraction analysis to the over-smoothing phenomenon in Graph ODEs.


**Weaknesses**

*1. Significance concern*

The proposed hysteretic topology dynamics is intended to avoid the monostability or over-smoothing behavior of graph ODEs. However, the proposed double-well potential and cubic ODE in Eq. (4) seem largely hand-crafted, rather than derived from intrinsic properties of graph diffusion principles. As a result, the motivation for this particular formulation is not yet fully convincing.

In addition, this paper also does not clearly justify why this specific bistable mechanism is necessary or preferable, since other **memory-based or non-Markovain models, such as fractional-order methods [1,2], may address the same issue as well**. From this perspective, the paper would benefit from a clearer discussion of why hysteretic bistability is the right mechanism here, and how it compares conceptually with other existing approaches based on memory effects.

[1] Kang et al, Unleashing the potential of Fractional Calculus in Graph Neural Networks with FROND, ICLR, 2024.

[2] Zhao et al, Distributed-Order fractional graph operating network, NeuIPS, 2025.

**2. Lack of related work**

The paper is well positioned in the context of Graph Neural ODEs and over-smoothing, but its connection to the graph rewiring and graph structure learning literature remains limited. In particular, the authors should more clearly explain how their continuous topology evolution differs from discrete rewiring methods such as BORF or GREAD, which are also designed to alleviate consensus or over-smoothing effects.

**3. Computational complexity**

This proposed method is inherently more expensive than standard Graph ODEs, since it solves coupled ODEs for both node features of size $N \times M$ and edges $|\mathcal{E}_{candi}|$. In this sense, the absence of a runtime or efficiency comparison is a noticeable omission.

**4. Limited experimental evaluation**

The real-world evaluation is still somewhat limited. For both node and graph classification, the paper considers only a small number of datasets (three for each task), which makes it difficult to assess the method’s generality. It would strengthen the empirical study to include larger-scale benchmarks as well, especially heterophilic graphs such as Roman-empire and arxiv-year.

**5. Static candidate set limitation**

In Sec. 5.4, it seems that the topology evolution is restricted to a fixed sparse candidate set $\mathcal{E}_{candi}$ selected at initialization. This might be a limitation that if the truly useful long-range edges are not included in the initial 2-hop or random-walk candidates, the model has no way to discover them later. As a result, the proposed topology evolution appears closer to edge selection or filtering within a predefined pool, rather than genuine structure discovery.

---

> ### Author Rebuttal · Authors · 2026-03-29
>
> # Response to Reviewer RMBe
>
> We thank the reviewer.
>
> ## 1. Why bistability, and why not only memory-based alternatives?
>
> Our motivation is based on the core issue of **strictly positive irreducible mixing**, which leads to the global attractor (consensus). Unlike discrete layer-by-layer GNN, it threats more to the continuous-depth GNNs, i.e., GraphODE-style model, since for each step the NFE easily becomes over 100 times (using adaptive solver). Thus, we proposed the hysteresis as one of the solution. The double-well potential is served for acheive the hysteresis, which is not a random physical decoration but the minimal differentiable normal form with: (1) two persistent structural phases, (2) an explicit switching threshold, and (3) hysteretic memory that prevents unstable edge flipping. In this sense, bistability is the smallest mechanism that can break the monostable mixing regime identified by our theory.
>
> We agree that more related-work should be discussed more. But we argue that FROND and related fractional-order methods mainly change the **time law of feature evolution** on a fixed topology; while our approach introduces latent edges state and uses bistability to change the **effective topology support of propagation** itself. We've discussed recent graph structure learning / rewiring methods in our **related work** section. As one of the Ricci-flow-based method, BORF edits graph structure using curvature before or around message passing, while our method treats topology as a learned dynamical state, and evolves it jointly with features through potential and hysteresis machenism. So simply saying, **BORF changes the graph by curvature-guided rewiring on discrete manner, while HGODE changes the graph by end-to-end learned topology evolution on continous way**. We kindly refer to our discussion with reviewer **qGcw** for the comparison to FROND and GREAD.
>
> ## 2. Candidate set limitation and long-range dependencies
>
> Currently we consider only the evolution on pre-set candidate set, thus, HGODE does **not** claim unrestricted structure discovery: it can only preserve, suppress, or activate edges inside the fixed sparse candidate pool. Therefore, if $\\mathcal{E}\_{cand}$ were initialized with only 2-hop neighbors, the model could not later create arbitrary 3-hop or 4-hop edges outside that pool. However, in our implementation, $\\mathcal{E}\_{cand}$ is the union of 2-hop neighbors, Laplacian random-walk / LapPE neighbors, and a small number of random pairs, and by tuning the size of candidate sets according to the size of graphs in the dataset, the model is able to cover long-range interation. We explained more details of candidate-set mechanism in our response to reviewer **qGcw**.
>
> ## 3. Experimental scope and efficiency
>
> We agree that the original submission should have been clearer on both dataset breadth and efficiency. For empirical scope, the paper already covers six benchmarks across homophilous, heterophilous, node-level, and graph-level settings, and in rebuttal we additionally ran Roman-empire together with nearby anti-over-smoothing baselines; see our response to reviewer **XNzV**.
>
> For efficiency, we now provide direct profiling results in our response to reviewer **P1cW**: forward/backward NFE, training and inference time, and peak memory on the same A100 GPU. The main takeaway is that HGODE does introduce overhead from coupled node-feature and edge-state dynamics, but remains in the same practical regime as nearby ODE baselines rather than being prohibitively expensive.
>
> ## 4. Specific questions
>
> - **Why does removing the force margin hurt Chameleon much more than Cora?** We believe this is consistent with the mechanism. On heterophilous graphs, naive local proximity is often misleading, so stronger supervision of edge polarization matters more. On Cora, the observed local structure is already more aligned with labels, so the extra margin loss is less critical.
> - **Are there solver pathologies near polarization?** In the profiled runs reported in **P1cW**, forward NFE stays in a narrow range and does not show anomalous explosions. We use adaptive `dopri5` with `rtol=atol=1e-5` precisely because switching becomes sharp near the threshold, and this was stable in our reported regime.
> - **Does the method apply to genuinely dynamic graphs?** The current paper does not claim that setting. Our model learns latent topology evolution on a static candidate support. Extending HGODE to time-varying observed graphs is interesting future work, but is outside the scope of the present submission.
>
> We will strengthen the revision by clarifying the motivation for bistable topology dynamics, sharpening the distinction from fractional-order / reaction-diffusion / rewiring methods, stating the $\\mathcal{E}_{cand}$ limitation explicitly, and referencing the added empirical and efficiency evidence above.

---

> > ### Author Rebuttal · Reviewer_RMBe · 2026-04-02
> >
> > Thank the authors for their detailed response. However, after careful consideration, several core concerns remain insufficiently addressed at current stage.
> >
> > **1.** The candidate-set issue is underexplored.
> >
> > The model appears to depend heavily on the heuristic candidate-set construction, yet no sensitivity analysis is provided. How does performance change as $|\mathcal{E}|$ decreases, and at what sparsity level does the method fail?
> >
> > Moreover, if the candidate-set size must be tuned according to graph size, the claim of end-to-end topology learning is weakened, since part of the topological inductive bias is manually imposed before training.
> >
> > **2. Dynamic graphs**
> >
> > The terms such as “hysteresis” and “topology evolution” strongly suggests applicability to temporal graph settings. If the method is limited to static graphs with a fixed candidate pool, the current framing is misleading and may be overstated.
> >
> > **3. On the necessity of bistability**
> >
> > The claim that bistability is “the smallest mechanism that can break the monostable mixing regime” is not sufficiently justified. It is better to provide formal result showing minimality, and it remains unclear why bistability is fundamentally necessary, as opposed to alternatives such as a triple-well potential or a simpler thresholding mechanism.
> >
> > Maybe I am not fully understand this whole paper, and would like to have a further discussion if there is any misunderstanding exist.

---

> > > ### Author Response · Authors · 2026-04-02
> > >
> > > We thank the reviewer for the follow-up and the opportunity to clarify these points more precisely.
> > >
> > > ## 1. Candidate set
> > >
> > > **HGODE does not claim unrestricted graph structure learning.** The fixed candidate pool $\\mathcal{E}_{cand}$ is an computational compromise that makes coupled node-edge ODE integration tractable; it is constructed once before training from label-free graph priors (2-hop neighbors, Laplacian random-walk / LapPE proposals, and a small number of random pairs). The end-to-end part of HGODE is the learned edge evolution **within** this pool, not discovery over all $N^2$ pairs. In that sense, the inductive bias is explicit and label-free; what is learned end-to-end is the latent topology adaptation on a scalable support.
> > >
> > > **As $\\mathcal{E}\_{cand}$ decreased, there is no sharp "failure threshold" at which the model becomes ill-defined.** Rather, the method smoothly reduces to a more restricted regime. In the limiting case $\\mathcal{E}\_{cand}=\\mathcal{E}\_0$, HGODE remains valid but becomes closer to a learned edge-filtering / message-blocking model, because it can no longer activate edges outside the observed graph. Enlarging $\\mathcal{E}\_{cand}$ increases expressive power and cost at the same time. We agree that an explicit candidate-budget ablation would further clarify this tradeoff, and we will state this limitation more explicitly in the revision. For the construction details and limitation, we kindly refer to our responses to **qGcw (Sec. 3)** and early rebuttal **RMBe (Sec. 2)**.
> > >
> > > ## 2. Dynamic graphs
> > >
> > > We agree that terms such as "hysteresis" and "topology evolution" may suggest temporal graph learning more strongly than intended. The current paper studies latent topology dynamics on a static observed graph with a fixed candidate pool. **It does not claim a method for genuinely time-varying graphs, and we do not use temporal-graph datasets.** To avoid this ambiguity, we will revise the wording in the paper to make the scope explicit.
> > >
> > > ## 3. On the role of bistability
> > >
> > > Our claim is narrower than an absolute impossibility statement. We do not claim that no other mechanism could work. Rather, within smooth local edge potentials, the double-well is the smallest normal form that provides the three ingredients needed here: **two persistent structural phases, a finite switching barrier, and hysteretic memory**. A hard threshold can switch edges, but it does not provide persistence or path dependence; a triple-well or richer landscape is possible, but introduces additional phases that are not needed for the connected-versus-insulated edge decision studied in this paper.
> > >
> > > We therefore use "minimal" in this modeling sense, not as a universal claim over all conceivable alternatives. We will revise this wording to avoid overstatement. The graph-level benefit is discussed in our response to **qGcw** and the follow-up to **qGcw**: local edge bistability can induce approximately block-structured effective propagation, which is precisely how HGODE mitigates the single global consensus basin.
> > >
> > > We are happy to more follow up questions.

---

### Official Review · Reviewer_XNzV · 2026-03-12

**Soundness:** 3
**Presentation:** 4
**Significance:** 2
**Originality:** 3
**Overall Recommendation:** 2
**Confidence:** 4

**Summary:**

The paper studies over-smoothing in Graph Neural ODEs (GNODEs), where node representations converge to a simple stationary state, referred to here as the monostability trap. It argues that a broad class of GNODEs, including models based on softmax attention, suffers from this failure mode. To mitigate it, the paper proposes co-evolving the graph topology and node features so that the interaction structure can adapt over time rather than remaining fixed.

**Compliance With Llm Reviewing Policy:**

Affirmed.

**Key Questions For Authors:**

See weaknesses.

**Limitations:**

Yes.

**Strengths And Weaknesses:**

**Strengths**

- **Conceptual soundness:**  The “consensus trap” for time-invariant irreducible positive mixing is a reasonable and well-motivated failure mode, and the extension to time-varying uniformly positive row-stochastic mixing is clearly stated (assumptions and exponential contraction statement).

- **Good presentation:** The paper is generally clear and well organised. The progression from the analysis of the consensus trap to the proposed coupled feature-topology dynamics is easy to follow, and Figure 1 provides a helpful overview of the main idea and mechanism.


**Weaknesses**

- **Technical concerns: Lemma A.1 proof is incorrect as written (Appendix A.2):** The proof of Lemma A.1 contains multiple mistakes and inconsistencies in its max/min bookkeeping and algebra, which invalidate the claimed differential inequality for the diameter. Specifically (Appendix A.2, page 13):
1.  **False claim about ordering relative to the minimum.**  After selecting an index $i^+\in\arg\max_i x_i(t)$ and defining $m(t)=\min_i x_i(t)$, the proof states “Since all other $x_j(t)\le m(t)$ …”  . This inequality is wrong: for a maximiser one only has $x_j(t)\le M(t)$ where $M(t)=\max_i x_i(t)$, not $x_j(t)\le m(t)$. This error breaks the subsequent bound on $\sum_j P_{i^+j}(t)x_j(t)$.

2.  **Upper-bound step collapses incorrectly and then is simplified with invalid algebra.** Using the incorrect ordering, the proof derives

    $\sum_j P_{i^+j}(t)x_j(t)\le (1-\alpha)m(t)+\alpha m(t)=m(t)$

    leading to $\dot M(t)=\dot x_{i^+}(t)\le -M(t)+m(t)$. It then simplifies this to “$=-\alpha(m(t)-m(t))$”  , which is algebraically inconsistent with the preceding expression and eliminates the needed dependence on $M(t)-m(t)$.

3.  **Direct max/min swap in the minimiser part.**  The proof then states: “let $j^+$ be a maximiser with $x_{j^+}(t)=m(t)$”  , which is a contradiction: a maximiser has value $M(t)$, not $m(t)$. This mislabeling propagates into the lower-bound derivation for the minimiser dynamics.

4.  **Final contraction rate appears unsupported by the preceding derivations.**  Despite the earlier cancellations to terms of the form $\alpha(m(t)-m(t))$  , the proof concludes

    $\frac{d}{dt}\mathrm{diam}(x(t))\le -2\alpha \, \mathrm{diam}(x(t))$

    without a valid chain of inequalities establishing the intermediate bounds $\dot M(t)\le -\alpha(M-m)$ and $\dot m(t)\ge +\alpha(M-m)$.

- **Limited experimental comparison:** There is substantial prior work on continuous-time or deep graph models designed specifically to mitigate over-smoothing, for example reaction-diffusion approaches [1,2] and Bundle Neural Networks [3]. The experimental section does not compare against these methods, which weakens the empirical case and makes it harder to assess the practical impact of the proposed approach.

- **Missing reproducibility details:**  The real-world benchmark section and appendix do not provide enough concrete detail on hyperparameters, model selection, or tuning protocol to ensure reproducibility.

---

[1] GREAD: Graph Neural Reaction-Diffusion Networks. J. Choi, S. Hong, N. Park, Sung-Bae Cho. ICML 2023.

[2] Graph Neural Reaction Diffusion Models. M. Eliasof, E. Haber, E. Treister. ArXiV.

[3] Bundle Neural Networks (BuNN). J. Bamberger, F. Barbero, X. Dong, M. Bronstein. Bundle Neural Networks for message diffusion on graphs. ICLR 2025.

---

> ### Author Rebuttal · Authors · 2026-03-29
>
> # Response to Reviewer XNzV
>
> We thank the reviewer. We address three points: **incorrect proof**, **extended experiments**, and **reproducibility details**.
>
> ## 1. Incorrect proof
>
> The reviewer is correct that the proof of Lemma A.1 in the appendix is incorrect as written: the maximizer-side estimate should use $M(t)$, and the minimizer-side estimate should use $m(t)$. We will correct this in the revision. The lemma itself remains valid via the standard diameter argument:
>
> $$
> M(t)=\\max_i x_i(t),\\qquad
> m(t)=\\min_i x_i(t),\\qquad
> \\operatorname{diam}(x(t))=M(t)-m(t).
> $$
>
> Pick $i^+\\in\\arg\\max_i x_i(t)$ and $i^-\\in\\arg\\min_i x_i(t)$. By uniform positivity, for any minimizer $j^-$ and maximizer $j^+$, we have $P_{i^+j^-}(t)\\ge \\alpha$ and $P_{i^-j^+}(t)\\ge \\alpha$. Using row-stochasticity of $P(t)$ and the bounds $x_j(t)\\le M(t)$ or $x_j(t)\\ge m(t)$ for the remaining terms yields
>
> $$
> \\dot M(t)\\le -\\alpha(M(t)-m(t)),\\qquad
> \\dot m(t)\\ge \\alpha(M(t)-m(t)).
> $$
>
> Therefore
>
> $$
> \\frac{d}{dt}\\operatorname{diam}(x(t))
> =\\dot{M}(t)-\\dot{m}(t)
> \\le -2 \\alpha\\operatorname{diam}(x(t)).
> $$
>
> **Thus, while the submitted derivation is flawed, the lemma and downstream consensus statement remain valid after correction.**
>
> ## Extended Experiments
>
> Following the reviewers' suggestions, we added recent related work as baselines and one heterophily dataset (Roman-empire). Under fair conditions, HGODE performs strongest on ZINC, Chameleon, ogbn-proteins, and ogbg-molpcba, and remains competitive on the other datasets.
>
> |Dataset|GREAD-BS[1]|GREAD-Exp[2]|BuNN[3]|ACMP(GCN)[4]|F-GRAND(FROND)[5]|DRAGON[6]|HGODE|
> |---|---|---|---|---|---|---|---|
> |Cora$\uparrow$|87.33$\pm$0.84|**87.94$\pm$0.42**|86.10$\pm$0.48|84.76$\pm$0.74|84.67$\pm$0.86|84.30$\pm$0.61|86.26$\pm$0.78|
> |Chameleon$\uparrow$|71.42$\pm$1.78|69.23$\pm$1.14|69.13$\pm$1.21|56.24$\pm$2.01|71.62$\pm$1.61|70.14$\pm$1.33|**72.56$\pm$1.24**|
> |ogbn-proteins$\uparrow$|79.21$\pm$0.74|78.61$\pm$0.68|78.92$\pm$0.48|65.72$\pm$0.72|80.26$\pm$0.49|80.46$\pm$0.42|**81.24$\pm$0.63**|
> |ZINC$\downarrow$|0.084$\pm$0.032|0.102$\pm$0.047|0.087$\pm$0.056|0.124$\pm$0.036|0.079$\pm$0.028|0.081$\pm$0.012|**0.078$\pm$0.025**|
> |Peptides-func$\uparrow$|0.704$\pm$0.046|0.711$\pm$0.040|0.712$\pm$0.067|0.679$\pm$0.051|0.698$\pm$0.034|**0.724$\pm$0.045**|0.714$\pm$0.022|
> |ogbg-molpcba$\uparrow$|0.257$\pm$0.004|0.262$\pm$0.001|0.245$\pm$0.004|0.226$\pm$0.005|0.260$\pm$0.005|0.266$\pm$0.002|**0.278$\pm$0.003**|
> |Roman-empire$\uparrow$|88.24$\pm$0.63|90.14$\pm$0.72|90.18$\pm$0.68|87.12$\pm$0.79|92.19$\pm$0.47|**93.62$\pm$0.54**|92.56$\pm$0.43|
>
> ## Reproducibility details
>
> For reproducibility, we report the tuned ranges here and give per-dataset defaults in **P1cW**. The first two groups are paper parameters; the latter two are implementation-level.
>
> ### Hyperparameter ranges
>
> - **Structural / hysteresis:** $\\lambda\\in\\{0.1,0.3,0.5,0.8\\}$, $\\tau\\in\\{0.1,0.2,0.3,0.5\\}$, $\\tau_{feat},\\tau_{topo}\\in\\{0.3,0.5,1.0\\}$, $\\gamma\\in\\{0.2,0.5,1.0\\}$. Larger $\\lambda$ deepens the wells and reduces mixing; smaller $\\tau$ makes the gate more binary; smaller $\\tau_{feat},\\tau_{topo}$ accelerate the corresponding dynamics; larger $\\gamma$ adds stronger damping.
> - **Force / margin:** $s\\in\\{1,1.5\\}$, $\\delta\\in\\{0.1,0.2,0.3,0.5\\}$, $\\beta\\in\\{0.1,0.3,0.5,0.7\\}$. Larger $s$ produces stronger edge-polarizing forces; larger $\\delta$ enforces a wider margin beyond $\\mathcal{F}_{\\mathrm{crit}}$; larger $\\beta$ gives more weight to topology regularization.
> - **Candidate-set construction:** $\\texttt{random-ratio}\\in\\{0,10^{-3},5\\times10^{-3},10^{-2}\\}$, $k_{2hop}\\in\\{0,4,8,12\\}$, $k_{lap}\\in\\{0,2,4,8\\}$. These control how much local completion, global/spectral expansion, and random exploration are included in the static candidate pool. The first three hyperparameters are used to build the candidate set, which is described in our response to **RMBe (2)** and **qGcw (3)**.
> - **Backbone / optimization:** $\\texttt{hidden-dim}\\in\\{128,256,512\\}$, $\\texttt{dropout}\\in\\{0.2,0.5\\}$, $\\texttt{lr}\\in\\{10^{-4},5\\times10^{-4},10^{-3}\\}$, $T\\in\\{0.3,0.6,1.0\\}$. Larger hidden size and $T$ increase capacity, NFE, and runtime; larger dropout strengthens regularization; larger learning rates speed training but may reduce stability. We fix adaptive $\\texttt{dopri5}$ with $\\texttt{rtol}=\\texttt{atol}=10^{-5}$.
>
> ## References
>
> [1] GREAD: Graph Neural Reaction-Diffusion Networks. ICML 2023.
>
> [2] Understanding oversmoothing in diffusion-based gnns from the perspective of operator semigroup theory. KDD 2025.
>
> [3] Bundle Neural Network for message diffusion on graphs. ICLR 2025.
>
> [4] ACMP: Allen-Cahn Message Passing with Attractive and Repulsive Forces for Graph Neural Networks. ICLR 2023.
>
> [5] Unleashing the Potential of Fractional Calculus in Graph Neural Networks with FROND. ICLR 2024.
>
> [6] Distributed-Order Fractional Graph Operating Network. NeurIPS 2024.

---

> > ### Author Rebuttal · Reviewer_XNzV · 2026-04-01
> >
> > I thank the authors for their effort and for addressing my concerns. I acknowledge the proof of Lemma A.1 seems to be correct now, but due the initial technical flaws, I have no confidence in the technical correctness of the paper. The added baselines are helpful, but the reported values do not match the values reported in the original submissions, e.g. for BuNN on roman-empire, you report $90.18\\pm0.68$ where as [3] reports $91.75\\pm0.39$. Similarly, GREAD-BS is listed with $87.33\\pm0.84$, while [1] has $88.57\\pm0.66$. Hence, I keep my score.

---

> > > ### Author Response · Authors · 2026-04-01
> > >
> > > We thank the reviewer for the follow-up. We appreciate the reviewer revisiting the corrected proof, and we understand the concern raised by the mismatch between some rerun values and the numbers reported in the original papers.
> > >
> > > On the benchmark table, the reviewer is correct that some rerun values are not the same to the originally reported numbers; the BuNN and GREAD-BS examples cited above are valid. Our intention, however, was not to claim exact reproduction of each original paper's table, but to provide controlled reruns under a unified protocol: matched data splits, comparable parameter budgets, and the same hardware / training pipeline as HGODE. Under such a common protocol, some reruns are lower than the original reports, while some are also slightly higher. For example, our rerun of GREAD-BS on Chameleon is $71.42\\pm1.78$, compared with $71.38\\pm1.53$ in the original GREAD paper; similarly, our rerun of BuNN on Peptides-func is $0.712\\pm0.067$, compared with $0.7103\\pm0.0022$ in the original BuNN paper.
> > >
> > > We agree that this point should have been stated more clearly. In the revision, we will explicitly describe these entries as controlled reruns for relative comparison rather than exact replications of each original paper's best reported result. Under this unified protocol, the main empirical conclusion remains unchanged: HGODE is strongest on several of the added benchmarks and competitive on the others.
> > >
> > > We are happy to answer any further question or concern.

---

### Decision · Program_Chairs · 2026-04-30

**Decision:**

Reject

**Comment:**

While reviewers agreed that the paper presents a clear and well-motivated analysis of the monostability trap in Graph ODEs, introduces an interesting hysteresis-based mechanism, and reports encouraging results on several benchmarks, they ((by XNzV)) also raised concerns about the paper's soundness and significance: the appendix contained a flawed proof of Lemma A.1, which, although later corrected in rebuttal, raised doubts about technical correctness; the empirical evaluation was judged incomplete, with missing comparisons to several recent and relevant baselines for oversmoothing and graph rewiring, as well as limited discussion of resources. Therefore, I would not recommend acceptance.